# Rectified Decoupled Dataset Distillation: A Closer Look for Fair and Comprehensive Evaluation

**Xinhao Zhong**[1]   **Shuoyang Sun**[1]   **Xulin Gu**[1]   **Chenyang Zhu**[3]
**Bin Chen**[1,2*]   **Yaowei Wang**[1,2]
[1]Harbin Institute of Technology, Shenzhen    [2]Peng Cheng Laboratory
[3]Tsinghua Shenzhen International Graduate School, Tsinghua University
xh021213@gmail.com, 24s151152@stu.hit.edu.cn,
210110720@stu.hit.edu.cn, chenyangzhu.cs@gmail.com,
chenbin2021@hit.edu.cn, wangyw@pcl.ac.cn;

## Abstract

Dataset distillation aims to generate compact synthetic datasets that enable models trained on them to achieve performance comparable to those trained on full real datasets, while substantially reducing storage and computational costs. Early bi-level optimization methods (e.g., MTT) have shown promising results on small-scale datasets, but their scalability is limited by high computational overhead. To address this limitation, recent decoupled dataset distillation methods (e.g., SRe$^2$L) separate the teacher model pre-training from the synthetic data generation process. These methods also introduce random data augmentation and epoch-wise soft labels during the post-evaluation phase to improve performance and generalization. However, existing decoupled distillation methods suffer from inconsistent post-evaluation protocols, which hinders progress in the field. In this work, we propose **R**ectified **D**ecoupled **D**ataset **D**istillation (RD$^3$), and systematically investigate how different post-evaluation settings affect test accuracy. We further examine whether the reported performance differences across existing methods reflect true methodological advances or stem from discrepancies in evaluation procedures. Our analysis reveals that much of the performance variation can be attributed to inconsistent evaluation rather than differences in the intrinsic quality of the synthetic data. In addition, we identify general strategies that improve the effectiveness of distilled datasets across settings. By establishing a standardized benchmark and rigorous evaluation protocol, RD$^3$ provides a foundation for fair and reproducible comparisons in future dataset distillation research. Our code is available at https://github.com/ndhg1213/RD3.

## 1 Introduction

Deep learning has rapidly advanced in recent years, with large-scale models trained on extensive datasets achieving impressive performance across diverse domains—most notably in computer vision He et al. (2016); Dosovitskiy et al. (2020) and natural language processing Devlin et al. (2018); Brown et al. (2020). However, training models on large-scale datasets typically incurs prohibitive computational and memory costs, posing significant challenges for deployment, especially in resource-constrained environments. Dataset distillation (DD) Wang et al. (2018) has emerged as a promising direction to address this issue by enabling the creation of compact synthetic datasets that retain the utility of the original data. Early information-matching methods Zhao & Bilen (2021b; 2023); Cazenavette et al. (2022) have achieved reliable performance on small-scale datasets Krizhevsky (2009), but their nested optimization structures imposed substantial time consumption, thereby limiting applicability to larger datasets Deng et al. (2009).

---

*Corresponding Author.

Recently, decoupled dataset distillation methods Yin et al. (2024); Su et al. (2024); Sun et al. (2024a) have been proposed to address this issue by separating model pre-training from data synthesis, significantly reducing computational costs. They further enhance performance by incorporating epoch-wise soft labels from teacher models during post-evaluation, achieving state-of-the-art results on large-scale benchmarks such as ImageNet-1K Deng et al. (2009). Existing decoupled approaches Yin et al. (2024) can be categorized into three paradigms based on their synthetic data generation mechanisms: optimization-based, selection-based, and generation-based methods. All these approaches share the common requirement of pre-training teacher models (either classifiers or generative models like diffusion models Song et al. (2021)) to achieve decoupling.

Specifically, optimization-based methods Yin et al. (2024); Shao et al. (2024a); Yin & Shen (2024); Du et al. (2024); Shao et al. (2024b) perform pixel-level optimization of synthetic datasets using pre-trained classifiers, guided by cross-entropy loss and Batch Normalization (BN) layer statistics. In contrast, selection-based methods Sun et al. (2024a); Zhong et al. (2024b) utilize classifiers or generative models to extract class-relevant visual regions directly from original images. On the other hand, generation-based methods Su et al. (2024); Gu et al. (2024) fine-tune generative models or optimize visual-textual embeddings to synthesize new images. Unfortunately, current research faces several significant challenges: First, inconsistent evaluation settings across various compression ratios, target datasets, and cross-architecture models pose substantial comparability barriers for researchers. Second, existing studies often overlook methodological commonalities, leading to

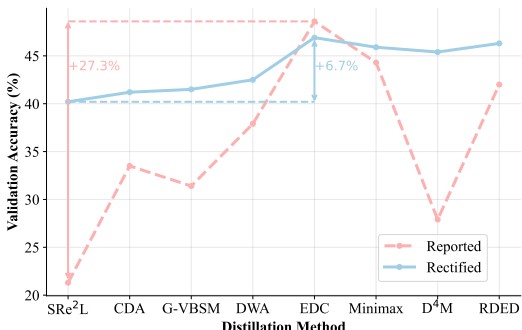

Figure 1: Performance comparison of various distillation methods evaluated by ResNet-18 on ImageNet-1K under IPC=10. Previous methods achieve a significant 27.3% performance improvement being influenced by multiple factors. After fairly reevaluating all methods under a unified setting, we obtained a rectified 6.7% performance enhancement.

incomplete comparisons that consider only subsets of the three aforementioned paradigms. More importantly, the inherent evaluation setting sensitivity in the post-evaluation phase results in performance comparisons being conducted under inconsistent settings, giving rise to confounded performance gains and significantly hindering the structured progress of this field. As shown in Figure 1, the performance gap reported by previous methods exceeds 27%. However, under unified and simplified settings, the actual improvements drop to less than 7%. This observation underscores a key challenge in dataset distillation: ***Claimed performance gains must be carefully disentangled to assess whether they arise from the core distillation mechanism or from auxiliary enhancements unrelated to the distillation process itself.***

To tackle these challenges, we introduce Rectified Decoupled Dataset Distillation (RD$^3$), a unified and comprehensive baseline framework under consistent post-evaluation settings that ensures fairness. Specifically, we conduct an in-depth investigation of the varied post-evaluation settings employed by prior methods, focusing on key parameters such as batch size and learning rate decay. Moreover, we establish a standardized evaluation protocol for decoupled dataset distillation, covering three critical dimensions: target datasets, compression ratios, and cross-architecture generalization. We then systematically replicate and re-evaluate the true performance and generalization capabilities of synthetic datasets generated by various methods. Our findings reveal that simply aligning evaluation settings suffices to eliminate substantial performance differences among synthetic datasets. The rectified results demonstrate that some reported performance gains primarily stem from improved post-evaluation settings rather than genuine enhancements in the quality of synthetic datasets.

Building upon RD$^3$, we highlight additional evaluation dimensions (e.g., time consumption) beyond test accuracy that are of greater importance. In addition, we identify several simple yet impactful techniques, such as using alternative initialization for optimization-based methods, that substantially influence test accuracy and may inadvertently introduce unfair advantages in future studies. This systematic exploration enables us to quantify and mitigate performance variations induced by implementation-specific modifications. To the best of our knowledge, this work represents ***the first exhaustive evaluation of representative decoupled dataset distillation methods under fully***

*standardized experimental conditions*. We anticipate that RD$^3$ will provide a robust foundation for meaningful comparisons and accelerate methodological advancements in this emerging field.

## 2 RELATED WORKS

### 2.1 BI-LEVEL DATASET DISTILLATION

Given a large-scale dataset $\mathcal{T} = \{(\mathbf{x}_i, y_i)\}_{i=1}^{|\mathcal{T}|}$, dataset distillation aims to generate a compact yet informative synthetic dataset $\mathcal{S} = \{(\mathbf{s}_i, y_i)\}_{i=1}^{|\mathcal{S}|}$, which preserves as much class-relevant information as possible while ensuring $|\mathcal{S}| \ll |\mathcal{T}|$. With $\mathcal{S}$, one can train a model from scratch with parameters $\theta$ :

$$\theta_{\mathcal{S}} = \arg\min_{\theta} \mathbb{E}_{(\mathbf{x},y) \in \mathcal{S}} \left( l(f_\theta(\mathbf{x}), y) \right). \tag{1}$$

where $l(\cdot, \cdot)$ represents the loss function and $f_\theta$ represents a classifier parameterized by $\theta$. Similarly, we define $\theta_{\mathcal{T}}$ for the original dataset $\mathcal{T}$. The primary objective can be formulated as:

$$\sup_{(\mathbf{x},y) \in \mathcal{T}} |l(f_{\theta_{\mathcal{T}}}(\mathbf{x}), y) - l(f_{\theta_{\mathcal{S}}}(\mathbf{x}), y)| \le \epsilon. \tag{2}$$

To achieve this, DD Wang et al. (2018) introduced a meta-learning approach based on a nested computation graph. However, the unrolled computation process incurs significant time costs. As an alternative, recent studies adopt a bi-level optimization framework that matches various proxy statistics between $\mathcal{S}$ and $\mathcal{T}$ formulated as:

$$\mathcal{S}^* = \arg\min_{\mathcal{S}} \mathcal{D}(f_{\theta'}(\mathcal{S}), f_{\theta'}(\mathcal{T})), \tag{3}$$

where $\mathcal{D}(\cdot, \cdot)$ represents different distance metrics used for matching, and $f_{\theta'}$ denotes the corresponding feature extractor. DC (Zhao et al., 2021) and DCC (Lee et al., 2022) minimize the distance between gradients in a progressively trained network, while DM (Zhao & Bilen, 2021a), CAFE (Wang et al., 2022), and DataDAM (Sajedi et al., 2023) focus on matching feature embeddings. Similarly, MTT (Cazenavette et al., 2022), TESLA (Cui et al., 2023), and DATM (Guo et al., 2023) align training trajectories to enhance learning consistency. Despite the significant achievement on small datasets (e.g., CIFAR10), bi-level distillation methods could not scale to the large-scale datasets (e.g., ImageNet-1K) due to prohibitive computational cost (Cui et al., 2022).

### 2.2 DECOUPLED DATASET DISTILLATION

Recent decoupled methods have significantly reduced computational complexity by decoupling the training processes of proxy models from synthetic dataset generation, while still achieving robust performance on large-scale datasets. Based on the different generation mechanisms, we categorize decoupled dataset distillation methods into three primary paradigms as follows.

**Optimization-based.** SRe$^2$L first introduced the decoupled optimization method by minimizing cross-entropy loss on synthetic datasets through pre-trained classifiers and aligning batch normalization (BN) statistics between synthetic and original datasets, which can be formulated as:

$$\mathbf{s}_i = \arg\min_{\mathbf{s}_i}[l(f_{\theta_{\mathcal{T}}}, \mathbf{s}_i) + \lambda \mathcal{L}_{\text{BN}}(f_{\theta_{\mathcal{T}}}, \mathbf{s}_i)], \tag{4}$$

where $\lambda$ denotes the weighting factor for the BN loss $\mathcal{L}_{\text{BN}}$. Building upon this, CDA Yin & Shen (2024) integrates curriculum learning into the optimization process and dynamically adjusts hyper-parameters during the post-evaluation phase. DWA Du et al. (2024) adopts real data initialization while further decomposing $\mathcal{L}_{\text{BN}}$, significantly enhancing the diversity of the synthetic dataset through pre-trained model perturbation. G-VBSM Shao et al. (2024a) utilizes multiple pre-trained models as teacher networks, simultaneously matching class-wise BN and convolutional statistics, and incorporates ensemble soft-labels and MSE loss during evaluation. EDC Shao et al. (2024b) further smooths the loss landscape in synthetic datasets and employs specialized evaluation-phase settings, positioning itself as the state-of-the-art (SOTA).

**Generation-based.** With the advancement of generative diffusion models, several methods have been developed to produce synthetic datasets by optimizing different components of the diffusion process. Minimax Gu et al. (2024) employs a DiT model Peebles & Xie (2023) pre-trained on ImageNet-1K,

fine-tuning it with a "minimax" criterion. Then the diffusion model is used to directly generate images. However, DiT has the limitation of only generating class label-conditioned images. As a result, Minimax cannot be applied to datasets that are Out-of-Distribution (OOD), such as CIFAR-10.

In contrast, $D^4M$ Su et al. (2024) uses Stable Diffusion (SD) pre-trained on LAION Schuhmann et al. (2022) as the backbone. It first performs k-means clustering on visual embeddings to obtain class centroids. These centroids are then combined with text prompts to generate synthetic datasets. $D^4M$ substantially enhances synthetic dataset diversity through SD and overcomes the OOD issue.

**Selection-based.** Subsequent studies have proposed generating synthetic datasets by identifying and cropping class-relevant visual regions, thereby reducing redundancy in large-scale datasets. RDED Sun et al. (2024a) performs random cropping on randomly sampled images and then ranks all patches in ascending order based on classification loss from a pre-trained classifier. Additionally, RDED concatenates multiple patches to form a single image to improve representativeness.

Subsequent methods have extended this paradigm by focusing on increasing the diversity of selected patches to improve generalization. FocusDD Hu et al. (2025) utilizes a pre-trained ViT Dosovitskiy et al. (2020) as the selector and incorporates class-irrelevant background patches. DPS Zhong et al. (2024b) employs SD as the selector, identifying class-relevant regions via differential text prompts (with and without class labels). Please refer to Appendix B for more detailed literature reviews.

### 2.3 EPOCH-WISE LABEL MATCHING

Early information-matching methods like MTT and subsequent improvements achieved superior performance using hard labels under extreme data compression scenarios, primarily applied to small-scale datasets. Recent studies Qin et al. (2024) suggest epoch-wise soft labels can better facilitate student model learning from synthetic datasets in large-scale settings, which can be formulated as:

$$\theta_{\mathcal{S}}^{t+1} = \arg\min_{\theta \in \Theta} L_{\text{KL}}(f_{\theta_{\mathcal{T}}}(\mathcal{A}(\mathcal{S})), f_{\theta_{\mathcal{S}}^t}(\mathcal{A}(\mathcal{S}))), \tag{5}$$

where $f_\theta^t$ represents the classifier at training epoch $t$, $\mathcal{A}(\cdot)$ denotes the random data augmentation, and $L_{\text{KL}}$ represents the Kullback–Leibler (KL) divergence. However, current distillation methods' epoch-wise soft-label implementations involve substantial misalignment: CDA employs smaller batch sizes, RDED utilizes a smoothed learning rate with stronger data augmentation, while G-VBSM and EDC generate hybrid soft labels through multiple teacher models. These implementation variances create significant obstacles for fair performance comparisons that urgently require resolution.

## 3 UNIFIED EVALUATION FRAMEWORK

We select well-known and state-of-the-art (SOTA) methods as baselines and categorize them into three groups: (1) Optimization-based methods Yin et al. (2024); Yin & Shen (2024); Shao et al. (2024a); Du et al. (2024); Su et al. (2024); (2) Generation-based methods Shao et al. (2024b); Gu et al. (2024); (3) Selection-based methods Sun et al. (2024a). Notably, although Minimax Gu et al. (2024) originally employs hard labels for evaluation, we include it in our consideration due to its applicability to large-scale datasets.

All subsequent methods adjust several evaluation-phase settings on the basis of $SRe^2L$, including (1) reduced training batchsize and increased training epochs, (2) carefully selected optimizer, (3) incorporation of extra loss function regularization, (4) hybrid soft labels , and (5) various data augmentations. Based on current knowledge, We are the first to make a thorough investigation across different methods. The difference with related works are shown in Appendix A.

### 3.1 DATASETS AND NETWORKS

**Datasets.** We adopt six standard image datasets: (1) CIFAR-10/100 Krizhevsky (2009), both of which have 50K 32×32 training images and 10K testing images from 10 and 100 classes. (2) ImageNet-1K Deng et al. (2009), consisting over 1,200,000 training images with various resolution from 1,000 classes. (3) TinyImageNet Le & Yang (2015), a subset of the ImageNet-1K with 200 classes. The training split contains 100K images, while the validation and test set include 10K images, All the images possess a resolution of 64×64. (4) ImageNette and ImageWoof Cazenavette et al. (2022), two

widely used coarse-grained and fine-grained subsets of ImageNet-1K, including 10 classes derived from ImageNet-1K. For all datasets, we conduct a comprehensive evaluation with IPC (image per class) from 1 to 100, which previous methods have not fully evaluated.

**Network Architectures.** We follow the settings used in previous works Yin et al. (2024); Sun et al. (2024a), employing ResNet-18 He et al. (2016) as the backbone network and applying soft labels across all settings. For the ResNet series, we additionally utilize ResNet-50/101 as more complicated evaluation models. For generalization evaluation, while employing CNN architectures like EfficientNet Tan & Le (2019) and MobileNet Howard (2017), we introduce Swin-T Liu et al. (2021) and ViT-B Dosovitskiy et al. (2020) from the ViT series as evaluation models, providing a timely and comprehensive assessment.

## 3.2 POST-EVALUATION SETTINGS

We adopt a standard setting with a single pre-trained ResNet-18 and KL divergence to generate soft labels and optimize the student model for simplification and fairness Yin et al. (2024); Yin & Shen (2024); Du et al. (2024); Sun et al. (2024a); Su et al. (2024). We summarize the unified settings in comparison to previous works and explain their motivations. Incremental post-evaluation impacts are presented in Figure 2, please refer to Appendix C for more detailed implementations and Appendix D for more results.

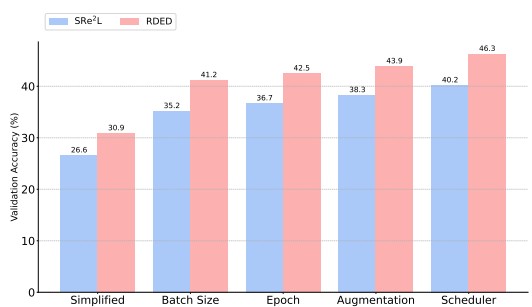

Figure 2: Performance comparison between SRe$^2$L and RDED on ImageNet-1K under IPC=10 evaluated by ResNet-18 with the same post-evaluation settings. The incremental techniques added from left to right lead to different performance impact.

**Training Epoch.** Early information-matching methods Zhao et al. (2021); Zhao & Bilen (2023); Cazenavette et al. (2022) typically train on synthetic datasets for around 1,000 epochs and evaluate the performance under overfitting, yet this setting is impractical for large-scale dataset applications. While most decoupled dataset distillation methods adopt 300 training epochs as widely used evaluation protocol, our preliminary experiments reveal that certain methods Sun et al. (2024a); Shao et al. (2024b) accelerate model convergence, which introduces bias in absolute performance comparisons. Therefore, we implement 400 training epochs during evaluation. As shown in Figure 2, the impact on performance remains minimal yet align the converge iterations.

**Batch size.** For few-shot learning tasks like dataset distillation, batch size (BS) exerts intriguing and significant impacts on experimental outcomes. SRe$^2$L employs BS=1024, while CDA uses BS=128 and demonstrates that smaller batch size yields notable improvements. Building upon this, RDED adopts varying BS sizes under different conditions. CV-DD Cui et al. (2025) further explore an extreme setting with BS=16. We further simplify and propose an optimized setting to balance the performance and efficiency: when evaluating synthetic datasets with ResNet-18, we uniformly set BS=50 across all settings unless $|S| < 50$, leading to nearly a 10% performance improvement across various methods as shown in Figure 2. For generalization tasks, we increase BS to 100 to mitigate gradient fluctuations induced by small batch sizes.

**Smoothing Learning Rate (LR) Scheduler.** For large-scale datasets like ImageNet-1K, existing methods employ Adam optimizer with an initial learning rate of 0.001 and a cosine annealing scheduler for optimization. RDED and EDC further implement smoothing LR scheduler to enhance the performance. Recent work CV-DD manually selects the scheduler smoothing factor $\zeta$ across different settings. Through preliminary experiments, we establish a universal yet competitive setting: using $\zeta = 1$ with ResNet-18 as evaluation model for finer-grained tuning as shown in Figure 2, while adopting $\zeta = 2$ for different architectures. Please refer to Appendix K for a intuitional comparison.

**Data Augmentation.** Under the premise of ensuring teacher-student model alignment through soft labels, previous methods universally enhanced synthetic dataset diversity via CutMix, Random Resized Cropped and Random Horizontal Flipped, achieving substantial performance gains. Building upon this, RDED and its optimized variant EDC introduce additional augmentation by exchanging patches with patch-concatenated images and expanding the crop ratio from 0.08 to 0.5. These

modifications collectively yield positive performance impacts as shown in Figure 2. Consequently, we adopt RDED's data augmentations as the universal standard and apply it across all methods. Detailed analysis are shown in Appendix M.

## 4 EXPERIMENTAL RESULTS AND ANALYSIS

### 4.1 DO METHODOLOGICAL DIFFERENCES TRANSLATE TO PERFORMANCE DISCREPANCIES?

Under the unified and fair settings provided by the $RD^3$ framework, we systematically reevaluated all decoupled dataset distillation methods, with the results presented in Table 1 and Table 2. Among optimization-based methods, EDC consistently demonstrates superior performance across all datasets and compression ratios, establishing itself as the representative method for this category. We subsequently compare EDC with generation-based and selection-based methods.

| Dataset | IPC | Optimization | | | | | Generation | | Selection |
| --- | --- | --- | --- | --- | --- | --- | --- | --- | --- |
| | | $SRe^2L$ | CDA | G-VBSM | DWA | EDC | Minimax | $D^4M$ | RDED |
| CIFAR10 | 1 | 16.2 ±0.7 | 16.4 ±0.6 | 17.5 ±0.7 | 18.3 ±0.3 | **26.6 ±0.5** | - | 13.4 ±0.8 | 22.5 ±0.7 |
| | 10 | 29.7 ±0.8 | 30.6 ±0.6 | 31.5 ±0.4 | 33.1 ±0.4 | **40.5 ±0.6** | - | 34.7 ±0.4 | 37.3 ±0.4 |
| | 50 | 53.9 ±0.5 | 54.5 ±0.7 | 55.6 ±0.4 | 59.9 ±0.4 | **64.8 ±0.3** | - | 61.9 ±0.4 | 63.3 ±0.2 |
| | 100 | 69.2 ±0.3 | 68.8 ±0.6 | 71.2 ±0.4 | 72.3 ±0.1 | 74.4 ±0.2 | - | **77.7 ±0.2** | 75.7 ±0.4 |
| CIFAR100 | 1 | 6.9 ±0.6 | 6.7 ±0.5 | 7.6 ±0.8 | 7.5 ±0.6 | **15.4 ±0.3** | - | 6.6 ±0.8 | 11.8 ±0.7 |
| | 10 | 32.6 ±0.5 | 33.5 ±0.5 | 38.9 ±0.6 | 41.3 ±0.5 | 46.6 ±0.4 | - | **47.8 ±0.5** | 44.4 ±0.5 |
| | 50 | 54.4 ±0.7 | 56.2 ±0.4 | 58.2 ±0.4 | 62.1 ±0.6 | **65.2 ±0.6** | - | 64.3 ±0.4 | 64.1 ±0.4 |
| | 100 | 59.6 ±0.4 | 60.7 ±0.3 | 63.3 ±0.4 | 64.2 ±0.4 | **69.1 ±0.5** | - | 68.9 ±0.2 | 67.5 ±0.2 |
| TinyImageNet | 1 | 6.1 ±0.8 | 7.1 ±0.5 | 6.2 ±0.3 | 6.8 ±0.8 | 10.2 ±0.6 | 9.8 ±0.7 | 3.9 ±0.8 | **11.1 ±0.9** |
| | 10 | 34.2 ±0.9 | 37.5 ±0.6 | 37.3 ±0.3 | 38.3 ±0.5 | 42.1 ±0.6 | 39.4 ±0.4 | 36.7 ±0.6 | **44.2 ±0.4** |
| | 50 | 52.5 ±0.7 | 53.0 ±0.6 | 53.7 ±0.6 | 54.2 ±0.3 | 57.1 ±0.4 | 54.4 ±0.4 | 53.8 ±0.4 | **58.7 ±0.4** |
| | 100 | 55.5 ±0.5 | 55.7 ±0.3 | 56.5 ±0.4 | 56.8 ±0.5 | 61.5 ±0.3 | 56.1 ±0.3 | 57.6 ±0.4 | **61.8 ±0.2** |
| ImageNette | 1 | 26.6 ±0.7 | 25.4 ±0.6 | 28.9 ±0.6 | 29.7 ±0.9 | **33.6 ±0.5** | 28.8 ±0.5 | 27.7 ±0.6 | 31.4 ±0.6 |
| | 10 | 56.7 ±0.6 | 54.6 ±0.4 | 61.6 ±0.4 | 64.3 ±0.4 | **70.6 ±0.4** | 66.6 ±0.5 | 66.3 ±0.5 | 63.8 ±0.5 |
| | 50 | 79.0 ±0.3 | 77.8 ±0.3 | 81.4 ±0.7 | 83.2 ±0.5 | 86.7 ±0.3 | 85.2 ±0.3 | 86.5 ±0.2 | **86.8 ±0.6** |
| | 100 | 85.2 ±0.2 | 84.7 ±0.5 | 87.7 ±0.3 | 87.1 ±0.1 | 90.3 ±0.4 | 89.3 ±0.2 | **90.7 ±0.1** | 89.6 ±0.4 |
| ImageWoof | 1 | 12.2 ±0.9 | 14.6 ±0.6 | 14.4 ±0.4 | 16.5 ±0.5 | **24.4 ±0.3** | 23.8 ±0.5 | 19.7 ±0.6 | 20.3 ±0.5 |
| | 10 | 26.8 ±0.5 | 25.7 ±0.5 | 34.5 ±0.5 | 36.1 ±0.5 | 42.3 ±0.6 | 45.5 ±0.6 | 35.4 ±0.5 | **46.5 ±0.6** |
| | 50 | 61.3 ±0.5 | 59.7 ±0.5 | 65.5 ±0.5 | 67.8 ±0.7 | **72.6 ±0.4** | 72.2 ±0.4 | 69.8 ±0.4 | 72.0 ±0.5 |
| | 100 | 69.5 ±0.4 | 68.7 ±0.4 | 71.4 ±0.5 | 75.2 ±0.8 | 79.3 ±0.2 | 79.2 ±0.1 | **80.3 ±0.3** | 78.6 ±0.4 |
| ImageNet-1K | 1 | 4.1 ±0.1 | 4.2 ± 0.8 | 4.2 ± 0.8 | 4.5 ± 0.9 | 7.0 ± 0.5 | 6.8 ± 0.5 | 5.4 ± 0.4 | **7.6 ± 0.5** |
| | 10 | 40.2 ± 0.3 | 41.2 ± 0.5 | 41.5 ± 0.6 | 42.5 ± 0.7 | **46.9 ± 0.6** | 45.9 ± 0.7 | 45.4 ± 0.6 | 46.3 ± 0.2 |
| | 50 | 55.2 ± 0.2 | 56.7 ± 0.6 | 56.6 ± 0.2 | 57.7 ± 0.5 | 60.1 ± 0.3 | **60.4 ± 0.2** | 60.2 ± 0.4 | 58.9 ± 0.7 |
| | 100 | 59.7 ± 0.4 | 60.6 ± 0.2 | 61.5 ± 0.4 | 62.1 ± 0.5 | 63.2 ± 0.1 | 62.2 ± 0.5 | **63.5 ± 0.2** | 61.5 ± 0.4 |

Table 1: Performance comparison across various datasets with well-known decoupled distillation methods. The highlight results denote the best performance achieved under different settings within our fair framework. "⌣" denotes the second performance, and "-" denotes the results could not obtained with certain settings.

On CIFAR-10/100, EDC achieves dominant performance advantages in most scenarios, underperforming $D^4M$ in only two specific compression ratio settings. However, the performance superiority diminishes when EDC is applied to higher-resolution datasets and more complex data domains. In contrast, Generation-based methods exhibit competitive performance on both ImageNette and ImageWoof, $D^4M$ particularly benefits from its image diversity advantages in large IPC settings. While Minimax maintains stable performance across all settings, it is limited by its label space and cannot be applied to datasets other than ImageNet-1K and its subsets. Notably, in a low IPC setting(e.g., IPC=1), $D^4M$ shows severe limitations, especially on the representative fine-grained dataset Image-Woof, while other methods maintain stable performance, showing that $D^4M$ cannot effectively condense class-relevant features under extreme settings.

The most challenging dataset ImageNet-1K reveals a distinct phenomenon: each of the four methods achieves the best performance under different IPC settings. RDED exhibits performance degradation with increasing IPC due to its limited diversity, mirroring trends observed in Image-Nette and Image-Woof. $D^4M$ and Minimax still demonstrate better scalability in high-IPC settings. Surprisingly, EDC maintains a competitive performance across all settings. We provide an additional qualitative analysis in Appendix O and visualizations in Appendix P.

| | | Optimization | | | | | Generation | | Selection |
|---|---|---|---|---|---|---|---|---|---|
| IPC | Rectified | SRe²L | CDA | G-VBSM | DWA | EDC | Minimax | D⁴M | RDED |
| 1 | - | - | - | - | - | **12.8 ± 0.1** | - | - | 6.6 ± 0.2 |
| | ✓ | 4.1 ±0.1 | 4.2 ± 0.8 | 4.2 ± 0.8 | 4.5 ± 0.9 | 7.0 ± 0.5 | 6.8 ± 0.3 | 5.4 ± 0.4 | **7.6 ± 0.5** |
| Δ | | - | - | - | - | (5.8 ↓) | - | - | (1.0 ↑) |
| 10 | - | 21.3 ± 0.6 | 33.5 ± 0.3 | 31.4 ± 0.5 | 37.9 ± 0.2 | **48.6 ± 0.3** | 44.3 ± 0.5 | 27.9 ± 0.0 | 42.0 ± 0.1 |
| | ✓ | 40.2 ± 0.3 | 41.2 ± 0.5 | 41.5 ± 0.6 | 42.5 ± 0.7 | **46.9 ± 0.6** | 45.9 ± 0.7 | 45.4 ± 0.6 | 46.3 ± 0.2 |
| Δ | | (18.9 ↑) | (7.7 ↑) | (10.1 ↑) | (4.6 ↑) | (1.5 ↓) | (1.6 ↑) | (17.5 ↑) | (4.3 ↑) |
| 50 | - | 46.8 ± 0.2 | 53.5 ± 0.3 | 51.8 ± 0.4 | 55.2 ± 0.2 | 58.0 ± 0.2 | **58.6 ± 0.3** | 55.2 ± 0.0 | 56.5 ± 0.1 |
| | ✓ | 55.2 ± 0.2 | 56.7 ± 0.6 | 55.7 ± 0.4 | 59.2 ± 0.3 | 60.1 ± 0.3 | **60.4 ± 0.2** | 60.2 ± 0.4 | 58.9 ± 0.7 |
| Δ | | (8.4 ↑) | (3.2 ↑) | (3.9 ↑) | (4.0 ↑) | (2.1 ↑) | (1.8 ↑) | (5.0 ↑) | (2.4 ↑) |
| 100 | - | 52.8 ± 0.3 | 58.0 ± 0.2 | 56.6 ± 0.2 | 57.7 ± 0.5 | - | - | **59.3 ± 0.0** | - |
| | ✓ | 59.7 ± 0.4 | 60.6 ± 0.2 | 61.5 ± 0.4 | 62.1 ± 0.5 | 63.2 ± 0.1 | 62.2 ± 0.5 | **63.5 ± 0.2** | 61.5 ± 0.4 |
| Δ | | (6.9 ↑) | (2.6 ↑) | (4.9 ↑) | (4.4 ↑) | - | - | (4.2 ↑) | - |

Table 2: Comparison of reported accuracy obtained from original papers and re-evaluated by RD³ on ImageNet-1K. "-" denotes the missing values in previous works.

*Summary: The observed performance differences among distillation methods are primarily attributable to inconsistencies in post-evaluation settings rather than inherent differences in data quality, and no individual method consistently outperforms the others.*

## 4.2 WHAT METRICS BEYOND TEST ACCURACY MATTER FOR EVALUATING QUALITY?

**Effectiveness vs Efficiency.** Another critical evaluation metric that has been systematically overlooked in previous studies is the time consumption for dataset generation. Existing performance comparisons remain incomplete due to their limitations within specific method categories (e.g., optimization-based). Given the minimal performance variations observed under our RD³ framework, efficiency emerges as a crucial evaluation criterion that needs comprehensive comparison.

For optimization-based and selection-based methods, total time consumption equals per-image processing cost multiplied by total image count. Generation-based methods require additional computation cost for diffusion model

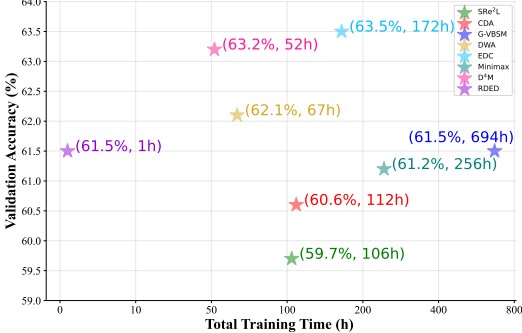

Figure 3: Comparison of the effectiveness and efficiency of all the decoupled distillation methods. Upper-left quadrant representing optimal effectiveness-efficiency balance.

fine-tuning Gu et al. (2024) or category centroid calculation Su et al. (2024) beyond basic generation time. To establish an intuitive and equitable efficiency comparison, we measured generation time under the most challenging IPC=100 setting. Notably, our evaluation excludes classifier training time required by optimization and selection methods, meaning their actual deployment costs would be substantially higher than the results we reported. All the experiments are conducted on a single Nvidia RTX-3090. The visualization shown in Figure 3 reveals that while performance differences remain marginal, time consumption varies by orders of magnitude (i.e., up to 100×).

**Generalization Ability.** To systematically investigate the intrinsic differences among different methods, we evaluate the corresponding synthetic datasets using diverse evaluation architectures. Unlike G-VBSM and EDC utilizing ensemble models to generate hybrid soft labels, we exclusively employ a single ResNet-18 for soft label generation to ensure maximum fairness in evaluation. Table 3 presents the experimental results of IPC=50 on ImageNet-1K. The performance variance across the ResNet family remains within 5%, with discrepancies decreasing as network depth increases.

| Method | ResNet-18 | ResNet-50 | ResNet-101 | MobileNet-V2 | EfficientNet-B0 | Swin-V2-T | ViT-B-16 |
|---|---|---|---|---|---|---|---|
| SRe$^2$L | 55.2 ± 0.2 | 62.8 ± 0.1 | 63.6 ± 0.4 | 48.1 ± 0.5 | 55.3 ± 0.4 | 55.3 ± 0.5 | 53.4 ± 0.3 |
| CDA | 56.7 ± 0.6 | 63.1 ± 0.2 | 64.2 ± 0.3 | 50.2 ± 0.2 | 56.0 ± 0.3 | 56.6 ± 0.3 | 53.9 ± 0.3 |
| G-VBSM | 56.6 ± 0.2 | 63.3 ± 0.4 | 63.8 ± 0.3 | 48.7 ± 0.1 | 56.1 ± 0.1 | 58.2 ± 0.7 | 57.8 ± 0.4 |
| DWA | 57.7 ± 0.5 | 63.3 ± 0.2 | 64.1 ± 0.4 | 52.1 ± 0.2 | 57.3 ± 0.4 | 57.9 ± 0.2 | 55.5 ± 0.5 |
| EDC | 60.1 ± 0.3 | **66.4 ± 0.3** | 66.0 ± 0.2 | 54.9 ± 0.2 | 59.6 ± 0.4 | **62.4 ± 0.3** | 61.6 ± 0.2 |
| Minimax | **60.4 ± 0.2** | 65.0 ± 0.3 | 64.6 ± 0.5 | 53.8 ± 0.1 | 59.9 ± 0.3 | 61.2 ± 0.3 | 62.3 ± 0.2 |
| D$^4$M | 60.2 ± 0.4 | 66.0 ± 0.3 | **66.5 ± 0.5** | **55.8 ± 0.2** | **61.4 ± 0.3** | 62.2 ± 0.3 | **63.7 ± 0.3** |
| RDED | 58.9 ± 0.7 | 65.2 ± 0.3 | 65.9 ± 0.2 | 53.5 ± 0.3 | 58.7 ± 0.4 | 61.3 ± 0.6 | 61.4 ± 0.3 |

Table 3: Generalization ability of synthetic dataset on ImageNet-1K under IPC=50. All the soft labels are generated by a single pre-trained ResNet-18 to ensure fairness.

| | SRe$^2$L | G-VBSM | DWA | EDC | RDED |
|---|---|---|---|---|---|
| Noise | 40.2 | 41.2 | **41.5** | 38.6 | 36.5 |
| Δ | - | - | - | (3.9 ↓) | (9.1 ↓) |
| Random | 41.8 | 42.6 | **46.4** | 42.5 | 45.6 |
| Δ | (1.6 ↑) | (1.4 ↑) | (4.9 ↑) | - | (1.3 ↓) |
| RDED | 41.5 | 42.0 | 46.1 | 43.1 | **46.9** |
| Δ | (1.3 ↑) | (0.8 ↑) | (4.5 ↑) | (0.6 ↑) | - |
| D$^4$M | 40.9 | 41.8 | 44.8 | 42.2 | **45.4** |
| Δ | (0.7 ↑) | (0.6 ↑) | (3.3 ↑) | (0.3 ↓) | (1.5 ↓) |

| | | SRe$^2$L | G-VBSM | EDC | D$^4$M | RDED |
|---|---|---|---|---|---|---|
| | - | 4.1 | 4.2 | 7.0 | 5.4 | **7.6** |
| IPC=1 | ✓ | 12.2 | 12.8 | 15.5 | 13.9 | **15.6** |
| Δ | | (8.1 ↑) | (8.6 ↑) | (8.5 ↑) | (8.5 ↑) | (8.0 ↑) |
| | - | 40.2 | 41.5 | **46.9** | 45.4 | 46.3 |
| IPC=10 | ✓ | 40.9 | 42.3 | **47.9** | 46.1 | 47.5 |
| Δ | | (0.7 ↑) | (0.8 ↑) | (1.0 ↑) | (0.7 ↑) | (1.2 ↑) |
| | - | 55.2 | 56.6 | 60.1 | **60.2** | 58.9 |
| IPC=50 | ✓ | 51.2 | 52.4 | **57.1** | 56.3 | 56.4 |
| Δ | | (4.0 ↓) | (4.2 ↓) | (3.0 ↓) | (3.9 ↓) | (2.5 ↓) |

Table 4: Performance comparison of optimization-based methods with different initialization on ImageNet-1K under IPC=10. ↓ and ↑ indicate the change direction of Δ compared to the default settings "-" of various distillation methods.

Table 5: Performance impact of using multiple teacher models to generate hybrid soft label on ImageNet-1K. The highlight results denote the best performance achieved across methods under different settings.

Performance divergences become more pronounced when testing other CNN-based models, with all methods achieving the poorest performance on MobileNet-B0.

Our extensive experiments conducted on transformer-based architectures reveal that most methods outperform the ResNet-18 baselines, demonstrating effective knowledge transfer. However, we observe that substantial performance degradation for SRe$^2$L, CDA, and DWA on ViT-B-16, potentially attributable to their limited image diversity, which hinders ViTs' learning capacity. In contrast, the superior diversity of D$^4$M enables it to achieve SOTA performance across most settings. More experimental results about generalization are shown in Appendix E and Appendix F.

*Summary: In comparison to marginal differences in test accuracy, computational efficiency should be considered a more critical criterion for evaluating different methods. Furthermore, generalization capability which often exhibits more pronounced variation offers a more informative metric.*

### 4.3 WHAT SUBTLE FACTORS INFLUENCE THE FIDELITY OF DISTILLED DATASETS?

**Alternative Initialization.** As thoroughly demonstrated in early information matching studies Zhao & Bilen (2021b); Liu et al. (2023), the initialization of synthetic datasets often plays a critical role in the field of dataset distillation. However, for existing optimization-based decoupled distillation methods, the use of superior initialization has become an underacknowledged practice. We systematically investigate the impact of initialization on optimization-based methods and explore potential combinations of different distillation methods by different initializations.

As shown in Table 4, different initializations exert substantial influence on specific methods. Notably, both DWA and EDC, which primarily aim to enhance dataset diversity, exhibit significant performance degradation when using Gaussian noise initialization.Conversely, G-VBSM demonstrates significant performance improvements when initialized with random sampling or RDED-generated images, occasionally outperforming EDC in certain settings. This is attributed to the inherent uncertainty introduced by its multi-teacher model matching mechanism during optimization. We provide a qualitative analysis in Appendix H.

**Hybrid Soft Label.** Despite G-VBSM and EDC emphasized that the involvement of multiple teacher models in image optimization necessitates the use of hybrid soft labels generated by these models during the relabel phase, the performance benefits of using hybrid soft label with other methods has not been explored. We implement hybrid soft labeling across five representative methods and evaluated them on ResNet-18. Experimental results shown in Table 5 reveal remarkable performance

| IPC | Loss | SRe²L | G-VBSM | EDC | D⁴M | RDED |
|---|---|---|---|---|---|---|
| 1 | KL | 4.1 | 4.2 | 7.0 | 5.4 | **7.6** |
|  | MSE-GT | 3.6↓ | 4.8↑ | 6.8↓ | 5.7↑ | **7.8**↑ |
| 10 | KL | 40.2 | 41.5 | **46.9** | 45.4 | 46.3 |
|  | MSE-GT | 40.9↑ | 42.3↑ | **47.9**↑ | 47.5↑ | 46.8↑ |
| 50 | KL | 55.2 | 56.6 | 60.1 | **60.2** | 58.9 |
|  | MSE-GT | 56.4↑ | 57.8↑ | 60.8↑ | **61.3**↑ | 60.2↑ |
| 100 | KL | 59.7 | 61.5 | 63.2 | **63.5** | 61.5 |
|  | MSE-GT | 59.5↓ | 61.9↑ | **64.1**↑ | 62.9↓ | 62.7↑ |

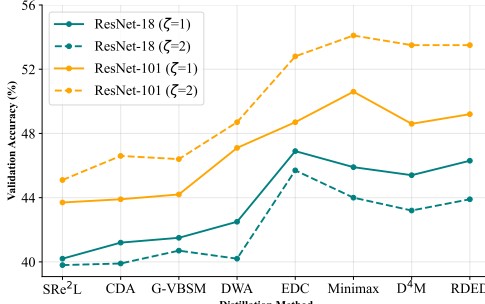

Table 6: Performance impact of using different loss functions on ImageNet-1K. ↓ and ↑ indicate the change direction compared to KL divergence.

Figure 4: Performance on ImageNet-1K under IPC=10 with different smoothing factor $\zeta$.

| Label | IPC | CIFAR10 | CIFAR100 | TinyImageNet | ImageNette | ImageWoof | ImageNet-1K |
|---|---|---|---|---|---|---|---|
| Hard | 1 | 16.4 ± 0.4 | 3.7 ± 0.6 | 1.9 ± 0.7 | 18.2 ± 0.5 | 10.2 ± 0.2 | 0.7 ± 0.4 |
|  | 10 | 23.5 ± 0.3 | 10.6 ± 0.4 | 4.1± 0.3 | 43.7± 0.4 | 23.2± 0.5 | 6.1± 0.6 |
|  | 50 | 31.6± 0.3 | 23.1± 0.3 | 10.1± 0.5 | 65.0± 0.4 | 35.5± 0.2 | 25.8± 0.3 |
|  | 100 | 40.6± 0.3 | 36.9± 0.5 | 17.2± 0.2 | 72.8± 0.2 | 41.8± 0.1 | 40.3± 0.1 |
| Soft | 1 | 25.5± 0.4 | 11.3± 0.6 | 6.6± 0.7 | 23.2± 0.7 | 16.3± 0.4 | 5.2± 0.5 |
|  | 10 | 42.3± 0.3 | 50.5± 0.2 | 39.3± 0.5 | 64.5± 0.3 | 36.5± 0.4 | 45.8± 0.1 |
|  | 50 | 66.4± 0.1 | 68.3± 0.2 | 57.3± 0.4 | 88.5± 0.2 | 69.3± 0.5 | 61.8± 0.2 |
|  | 100 | 80.1± 0.5 | 70.9± 0.2 | 59.9± 0.2 | 90.8± 0.1 | 74.8± 0.3 | 64.1± 0.2 |

Table 7: Performance of random sampling with hard label and soft label. The random images show a strong performance especially with soft label.

gains, SRe²L achieve a staggering twofold improvement with hybrid labels under IPC=1, while RDED and D⁴M also demonstrate significant enhancements even though their generation processes do not involve any other teacher models. Although the improvement magnitude decrease at IPC=10, consistent performance gains were still observed. The full results are shown in Appendix I. This evidence confirms that hybrid soft labeling is a universal and impactful technique.

*Summary: There exist general techniques aimed at refining the quality of synthetic datasets along two dimensions (i.e., the images and soft labels), which can lead to substantial performance variations without changing existing methods.*

### 4.4 WHICH OVERLOOKED VARIABLES UNDERMINE FAIR EVALUATION IN EVALUATION?

**Optimization Objective.** We investigate the impact of another previously misaligned loss function selection across all methods. G-VBSM replaces the commonly used KL divergence $D_{\text{KL}}(\cdot||\cdot)$ in other decoupled distillation methods Yin et al. (2024); Yin & Shen (2024); Su et al. (2024) with MSE + $\gamma \times$ GT (i.e., combining mean squared error and ground truth alignment). This modification draws from two insights: (1) The theoretical perspective proposed in Kim et al. (2021) that KL divergence becomes equivalent to MSE as $\tau \to \infty$. (2) The standard knowledge distillation practice of incorporating ground truth alignment as regularization.

To eliminate confounding variables, we systematically apply different loss function to all distillation methods. We set $\gamma$=0.025, which is a value empirically validated as acceptable in G-VBSM's original paper. Experimental results shown in Table 6 demonstrate that under our RD³ framework, while the new loss function does not guarantee consistent performance gains, it produces positive effects in most settings. Moreover, these improvements could potentially be amplified through optimal $\gamma$ selection. A comprehensive results are provided in Appendix J and we provide the experimental results under hard label setting in Appendix G and diverse knoledge distillation techniques in Appendix N. This finding suggests that the impact of loss function requires proper ablation studies when comparing against baselines in future research.

**Optimization Scheduler.** SRe²L first demonstrated that employing Adam optimizer with cosine annealing on large datasets enhances stability and performance. Building upon this foundation, RDED and EDC adopted smoothing learning rate. The mathematical formulation of this schedule is given by $\eta_i = \frac{1+\cos(\pi i/\zeta N)}{2}$, where $\eta_i$ represents the learning rate at epoch $i$ and $N$ represents the

total epoch number. However, recent work CV-DD Cui et al. (2025) achieved superior performance through manual adjustment of $\zeta$ across different datasets, compression ratios, and evaluation model architectures.

To isolate the influence of $\zeta$, we first evaluate all methods using both ResNet-18 and ResNet-101 as evaluation models. Our analysis visualized in Figure 4 reveals that with identical teacher and student models, faster learning rate decay facilitates earlier entry into fine-tuning phases. While evaluating with larger models (e.g., ViT-B-16), excessively small learning rates often hinder effective learning, resulting in an unconverged solution at the end of optimization.

*Summary: Consistent performance improvements can be achieved solely through adjustments to the training protocol, even when the synthetic dataset is held fixed. To ensure fairness, all subsequent methods should adopt a unified training setting, regardless of their specific motivations.*

## 5 WHAT CONSTITUTES THE OVERLOOKED BASELINE IN PREVIOUS COMPARISON?

Compared to traditional information-matching optimization methods, a significant performance gain in decoupled dataset distillation methods arises from their use of multi-round soft labels. To investigate the true efficacy of this method, we explored constructing the generated dataset using directly sampled random images.

Experimental results as shown in Table 7 reveal a surprising observation: under the soft label paradigm, simple random sampling outperforms all existing decoupled dataset distillation methods on CIFAR-10/100, ImageNette, and ImageNet-1K. This outcome aligns with findings in existing literature Xiao et al. (2025). Further analysis demonstrates that for coarse-grained datasets like ImageNette and ImageNet-1K, randomly sampled images maximize diversity while maintaining alignment with the teacher model's learned knowledge, thereby achieving strong soft label consistency. Conversely, on fine-grained datasets such as TinyImageNet and ImageWoof, existing decoupled methods excel by generating highly representative images that facilitate student model learning, whereas random sampling often introduces ambiguous patterns that degrade performance.

Under the hard label setting, Minimax and RDED surpass random sampling across most datasets and compression ratios. This advantage stems from their ability to produce realistic and representative images that align with category distributions, thereby aiding student model training. In contrast, optimization-based methods and $D^4M$, lacking explicit knowledge alignment via soft labels from teacher models, generate images with weak correlations to hard labels. This limitation severely hinders student models from learning accurate representations, resulting in performance far inferior to random sampling.

*The field of decoupled dataset distillation has historically overlooked the fact that randomly sampled images with soft labels constitute a powerful baseline, which in some cases can even outperform all existing distillation methods. Future work should pay greater attention to this phenomenon and adopt random sampling as a strong comparative benchmark.*

## 6 CONCLUSION

In this work, we revisit common inconsistencies in experimental settings used to compare decoupled dataset distillation methods and highlight the importance of establishing fair and comprehensive evaluation protocols. To this end, we introduce $RD^3$, a systematic re-evaluation framework that distinguishes true methodological improvements from performance gains driven by favorable hyper-parameter tuning. Our empirical analysis reveals that many reported advances are largely attributable to hyperparameter optimization rather than substantive algorithmic innovations. Building on these insights, we further investigate the prevalence of evaluation inconsistencies and provide refined performance assessments. Our findings offer actionable guidance for future work aimed at genuinely improving the quality of synthetic datasets.

**Acknowledgement.** This work is supported in part by the National Natural Science Foundation of China under grant 62576122, 62301189, and Shenzhen Science and Technology Program under Grant KJZD20240903103702004.

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

APPENDIX

## A    COMPARISON WITH EXISTING EVALUATION WORKS

### A.1    DC-BENCH

At the time of its release, DC-BENCH Cui et al. (2022) did not include decoupled data distillation methods, as such methods had not yet been proposed. Consequently, the benchmark only covered basic bi-level distillation methods. Moreover, due to their substantial computational and memory requirements, these methods are not scalable to large datasets or higher IPC settings, limiting the coverage and applicability of DC-BENCH compared to our work.

### A.2    PCA

PCA Xiao et al. (2025) re-evaluates existing optimization-based decoupled distillation methods under the CDA setting on ImageNet-1K, revealing that their performance often falls below that of random sampling and proposing data adjustment strategies to address this gap. In contrast, our work establishes a more comprehensive definition of decoupled distillation methods, explicitly incorporating generation-based methods. We further demonstrate that existing decoupled distillation methods do not consistently underperform random sampling across all datasets, especially on fine-grained datasets (e.g., ImageNet-Woof). Additionally, our study provides a deeper analysis of how various evaluation settings influence the performance of all types of decoupled distillation methods, offering insights that inform future improvements.

### A.3    DD-RANKING

DD-Ranking Li et al. (2024) introduces a new evaluation metric by computing accuracy gaps between distilled and randomly sampled datasets under different configurations to unify the evaluation of both bi-level and decoupled distillation methods. However, it does not offer a standardized benchmark framework nor investigate the performance discrepancies among different synthetic datasets. While our method make an in-depth analysis on how the various settings influence the test accuracy of different decoupled distillation methods.

## B    LITERATURE REVIEW

### B.1    $SR^2L$

$SRe^2L$ (Yin et al., 2024) first proposed the decoupled concept, drawing method from data-free knowledge distillation Yin et al. (2020) to completely disentangle proxy model training from data optimization processes, thereby reducing the substantial time overhead required by traditional information-matching based optimization methods Wang et al. (2018); Zhao & Bilen (2021a;b); Cazenavette et al. (2022); Wang et al. (2022); Shang et al. (2024); Zhong et al. (2024a). Since $SRe^2L$ inherits the data-free distillation framework, it employs Gaussian noise initialization which poses significant challenges for optimizing towards real data distribution. To address this, $SRe^2L$ simultaneously aligns dynamic BN statistics from synthetic datasets with teacher network's frozen BN information during optimization, thereby further constraining the generated dataset's distribution. Additionally, to resolve parameter dependency issues caused by single proxy model usage, $SRe^2L$ pioneered the introduction of epoch-wise soft labels during student model training to maximize knowledge transfer and alignment. The method further incorporates data augmentation operations like CutMix and RandomResizedCrop during alignment phases to enhance dataset diversity and boost performance. Subsequent methods have expanded the pipeline to object detection (Qi et al., 2024) and video classification (Gu et al., 2025) tasks.

### B.2    CDA

Building upon $SRe^2L$'s foundation, CDA Yin & Shen (2024) introduces curriculum learning into the image optimization process by implementing adaptive progressive cropping from small to large

scales in generated datasets, thereby achieving difficulty-graduated optimization schemes. This framework further modifies hyperparameters in SRe$^2$L's training evaluation procedures. Specifically, reducing BS yields substantial performance improvements. However, CDA regrettably omits thorough experimental analysis of these setting modifications while failing to isolate and validate the actual performance contributions from the curriculum learning component through ablation studies.

## B.3  G-VBSM

Although SRe$^2$L significantly enhances student model performance by utilizing teacher-generated soft labels during the relabel phase for knowledge transfer, it inadvertently causes generated datasets to overfit to single teacher parameters, thereby compromising generalization capability. To address this limitation, G-VBSM Shao et al. (2024a) introduces model pools comprising diverse architectures to optimize generated datasets, effectively reducing dependency on specific parameters and architectural settings. The method extends alignment objectives beyond Batch Normalization statistics to include convolutional features during optimization, while shifting the optimization scale from IPC to category-level for enhanced intra-class data diversity. G-VBSM further proposes more effective loss functions during evaluation phases to constrain information-rich datasets, explicitly requiring soft labels to be generated through collaborative predictions from architecturally heterogeneous teacher models. Notably, while maintaining SRe$^2$L's original hyperparameter settings (e.g., batch size) in post-evaluation phases, G-VBSM's novel settings remain untested on datasets generated by SRe$^2$L, leaving unresolved whether these modifications specifically cater to its own optimization characteristics. And the additional matching strategy introduced during recover phase leads to ten times time consumption.

## B.4  DWA

Since SRe$^2$L employs Gaussian noise initialization for generated datasets, the optimization process must rely on the mean values in BN statistics to approximate the original data distribution, which severely compromises the diversity of generated datasets. DWA Du et al. (2024) initially samples from the original dataset as initialization, then decouples the mean and variance components in the BN-based loss function while allocating greater optimization weights to the variance component. Subsequently, it introduces weight perturbations to teacher models to further enhance dataset diversity. Notably, DWA not only achieves substantial performance improvements by adopting CDA's parameter settings, but also significantly boosts computational efficiency through true initialization that accelerates optimization convergence. Using better initialization has consequently emerged as a simple yet effective performance enhancement technique in subsequent research.

## B.5  EDC

Building upon G-VBSM, EDC Shao et al. (2024b) introduces systematic improvements across three critical phases. During dataset generation, EDC advances beyond DWA's random sampling initialization by employing RDED-generated images as starting points. This strategic initialization effectively constrains redundant degrees of freedom arising from multi-teacher collaborative optimization while dramatically accelerating convergence. The framework innovatively incorporates flatness regularization through rigorous analysis of loss landscapes during optimization, achieving sharpness-aware minimization. For relabeling phases, EDC implements refined settings including reduced batch sizes as the same as RDED did and enhanced teacher model selection for improved soft label blending. The changes during evaluation stage include Further batch size reduction, precision-tuned learning rate schedulers, and EMA-based assessment mechanisms for performance refinement. Despite achieving multi-fold performance gains in specific settings, EDC critically overlooks two crucial aspects: (1) Systematic verification of proposed techniques' generalizability beyond distillation contexts. (2) Failure to disentangle performance improvements between dataset quality and evaluation protocol enhancements. This methodological gap exacerbates existing inconsistencies in decoupled dataset distillation frameworks, where performance metrics become confounded by optimized evaluation hyperparameters.

### B.6 MINIMAX

With rapid advancements in diffusion models, these architectures have been successfully integrated into dataset distillation frameworks. Unlike conventional parametric distillation approaches that employ GANs Cazenavette et al. (2023); Zhong et al. (2025), diffusion-based methods directly generate images through learned stochastic processes rather than pixel-level optimization, simultaneously enhancing generalization capabilities and reducing computational overhead. Minimax Gu et al. (2024) utilizes DiT pretrained on ImageNet-1K as foundational models, implementing a novel regularization strategy that expands feature distances to the most similar samples while contracting distances to dissimilar counterparts during diffusion model fine-tuning. This optimization ensures generated samples effectively approximate the original dataset distribution. Although Minimax incorporates CutMix for performance enhancement, it intentionally omits epoch-wise soft labels. Given its demonstrated scalability to large-scale datasets, we formally categorize Minimax within the decoupled dataset distillation and conduct comprehensive performance re-evaluation with soft label.

### B.7 D$^4$M

While Minimax demonstrates notable performance on ImageNet and its subsets, the diffusion model fine-tuning process incurs substantial temporal costs. This method employs DiT models and relies on one-hot labels as categorical prompts, fundamentally restricting its applicability to other datasets like CIFAR-10/100. D$^4$M Su et al. (2024) addresses these constraints by building upon Stable Diffusion (SD), a text-to-image generation diffusion model. The D$^4$M pipeline processes datasets through VAE encoders to obtain visual embeddings, conducts latent space clustering for class centroid derivation, and finally synthesizes images by combining these centroids with corresponding textual prompts. Although D$^4$M surpasses SRe$^2$L in generating semantically coherent images through diffusion mechanisms, its performance remains inherently dependent on the generation-based model's capabilities. The unconstrained visual embeddings frequently deviate from SD's latent data distribution, resulting in category-irrelevant image generation. While such anomalies may enhance dataset diversity when employing soft labels, they significantly impair performance on fine-grained datasets where precise feature representation is crucial, ultimately leading to accuracy degradation.

### B.8 RDED

For high-resolution datasets such as ImageNet-1K and its subsets, RDED Sun et al. (2024a) initially performs random cropping on original images and subsequently employs a pre-trained classifier to score and rank patches based on loss magnitude, ultimately stitching multiple high-scoring patches into composite images. In contrast, for low-resolution datasets like CIFAR-10/100, the framework directly scores and sorts original images through classifier evaluation while omitting cropping and concatenation operations. Distinct from alternative methods, RDED achieves remarkable computational efficiency by eliminating the training process entirely, with its synthesized datasets maintaining central positioning within the original data distribution. The framework further enhances synthetic dataset performance through implementation of reduced batch sizes and optimized learning rate decay schedules, demonstrating superior adaptability across varying resolution domains.

### B.9 EMERGING METHODS

Recent advancements in decoupled dataset distillation continue to emerge with notable methodological innovations. Here, we briefly summarize the subsequent methods.

DELT Shen & Xing (2022) addresses the trade-off between intra-class diversity and representational fidelity inherent in optimization-based approaches by initializing with RDED-generated datasets and selectively optimizing partial samples during training, thereby achieving enhanced performance with improved efficiency.

CV-DD Cui et al. (2025) employs multi-teacher model classification losses for joint optimization of synthetic datasets while establishing a strengthened baseline through our proposed universal techniques integrated with SRe$^2$L framework. Regrettably, these enhancements remain absent in other baseline implementations, leading to suboptimal solutions. In generation-based approaches,

IGD Chen et al. (2025) leverages DiT with influence function-guided optimization to amplify dataset representativeness, complemented by gradient-informed strategies for diversity augmentation. Regarding selection-based methods,

DDPS Zhong et al. (2024b) identifies RDED's classification model-driven evaluation as severely compromising diversity, instead adopting diffusion model-guided loss differentials calculated with text prompt under labeled and unlabeled conditions to localize class-relevant regions.

FocusDD Hu et al. (2025) utilizes pre-trained ViT as patch extractors with attention-driven visual saliency mapping, while incorporating irrelevant background images to further diversify synthetic datasets.

MGD$^3$ Chan-Santiago et al. (2025) introduces a plugin for diffusion model inference that guides the denoising direction with mode signals, encouraging the generation process to focus on more class-informative and prominent regions. During evaluation, MGD$^3$ adopts hard-label settings from Minimax and soft-label settings from D$^4$M.

Although these methods collectively advance dataset distillation effectiveness, their comparative analyses frequently neglect baseline setting alignment and essential ablation studies, thereby accentuating the critical necessity of our proposed systematic evaluation framework.

## C    IMPLEMENTATION DETAILS

Since the generation process of different methods is extremely different, we do not report the corresponding hyper-parameters for a simplified version. Overall, the re-generation of synthetic dataset follows the consistent settings of previous works. The only variations occur during post-evaluation phase, and we list the implementation details as follow.

| Implementation Details for Post-Evaluation on ResNet-18 | |
| --- | --- |
| Optimizer | Adamw |
| Learning Rate | 0.001 |
| Loss Function | KL-Divergence |
| Batch Size | 50 or $|\mathcal{S}|$ ($|\mathcal{S}| < 50$) |
| Epochs | 400 |
| Scheduler | Cosine Annealing |
| Smoothing Factor | $\zeta = 1$ |
| Augmentation | PatchShuffle, RandomResizedCrop, Horizontal Flip, CutMix |

Table 8: Hyperparameters for post-evaluation on ResNet-18 across various datasets.

| Implementation Details for Post-Evaluation on Other Architectures | |
| --- | --- |
| Optimizer | Adamw |
| Learning Rate | 0.001 |
| Loss Function | KL-Divergence |
| Batch Size | 100 or $|\mathcal{S}|$ ($|\mathcal{S}| < 100$) |
| Epochs | 400 |
| Scheduler | Cosine Annealing |
| Smoothing Factor | $\zeta = 2$ |
| Augmentation | PatchShuffle, RandomResizedCrop, Horizontal Flip, CutMix |

Table 9: Hyperparameters for post-evaluation task on other architectures across various datasets..

### C.1    RESNET-18

For ResNet-18, since the teacher and student models share identical architectures, specific hyperparameters must be employed during training to achieve optimal performance. As shown in Table 8, using smaller BS enables the student model to acquire more precise knowledge through soft labels. Simultaneously, employing smaller $\zeta$ values accelerates learning rate decay, allowing the student model to enter fine-tuning phases faster for acquiring refined knowledge.

### C.2    CROSS ARCHITECTURE

For other architectural settings where significant disparities exist between teacher and student models, particularly for ViT-based architectures that are substantially larger than ResNet-18, knowledge alignment through soft labels often proves challenging. Therefore, two complementary strategies are required as shown in Table 9. Larger batch sizes mitigate gradient fluctuation effects to better approximate the original dataset distribution. Larger $\zeta$ values maintain learning rates at higher ranges during initial phases to facilitate effective convergence learning.

## C.3 Different Evaluation Settings introduced by Previous Methods

For a clear comparison, we summarize the post-evaluation settings used for each method in Table 10. This provides strong support for the necessity of our work. We acknowledge that adjusting evaluation settings may lead to improved performance, and we encourage future work to design task-specific enhancement techniques, it need to be emphasized that any changes made during the post-evaluation phase must be tested across all baselines to assess their true impact on performance.

| Config | SRe$^2$L | CDA | G-VBSM | DWA | EDC | Minimax | D$^4$M | RDED |
|---|---|---|---|---|---|---|---|---|
| Label | Soft | Soft | Hybrid Soft | Soft | Hybrid Soft | Soft | Soft | Soft |
| Loss | KL | KL | MSE-GT | KL | MSE-GT | CE | KL | KL |
| Batchsize | 1024 | 128 | 1024 | 128 | 100 | 256 | 1024 | 100 |
| LRS ($\zeta$) | 1 | 1 | 1 | 1 | 2 | 2 | 1 | 2 |
| Data Augmentation | CutMix | CutMix | CutMix | CutMix | CutMix + Patch Shuffle | CutMix | CutMix | CutMix + Patch Shuffle |

Table 10: Different evaluation settings introduced by previous methods. The genuine quality improvement is conflicted by unaligned settings.

| Model | IPC | ImageNet-1K | | | | | | | |
|---|---|---|---|---|---|---|---|---|---|
| | | Optimization | | | | | Generation | | Selection |
| | | SRe$^2$L | CDA | G-VBSM | DWA | EDC | Minimax | D$^4$M | RDED |
| ResNet-50 | 1 | 4.7 ±0.2 | 4.3 ± 0.1 | 4.6 ± 0.2 | 4.8 ± 0.4 | 7.3 ± 0.3 | 7.7 ± 0.3 | 6.2 ± 0.4 | **8.2 ± 0.3** |
| | 10 | 48.5 ± 0.4 | 49.2 ± 0.3 | 49.5 ± 0.2 | 50.1 ± 0.4 | 53.9 ± 0.2 | **54.1 ± 0.2** | 53.3 ± 0.3 | 53.2 ± 0.2 |
| | 50 | 62.8 ± 0.2 | 63.1 ± 0.5 | 63.3 ± 0.3 | 63.3 ± 0.2 | 65.2 ± 0.2 | 65.0 ± 0.1 | **66.0 ± 0.2** | 65.2 ± 0.1 |
| | 100 | 64.9 ± 0.4 | 65.2 ± 0.2 | 65.5 ± 0.1 | 65.6 ± 0.5 | 66.9 ± 0.1 | 67.1 ± 0.2 | **67.4 ± 0.2** | 66.9 ± 0.4 |
| ResNet-101 | 1 | 3.4 ± 0.2 | 3.8 ± 0.1 | 3.6 ± 0.2 | 4.0 ± 0.4 | 6.2 ± 0.3 | 6.0 ± 0.3 | 4.7 ± 0.4 | **6.8 ± 0.3** |
| | 10 | 45.1 ± 0.2 | 49.6 ± 0.2 | 49.4 ± 0.4 | 48.7 ± 0.4 | 52.8 ± 0.2 | **54.8 ± 0.2** | 53.5 ± 0.4 | 53.6 ± 0.2 |
| | 50 | 63.6 ± 0.2 | 64.2 ± 0.1 | 63.8 ± 0.4 | 64.1 ± 0.4 | 66.0 ± 0.5 | 65.6 ± 0.2 | **66.5 ± 0.4** | 65.9 ± 0.3 |
| | 100 | 65.6 ± 0.1 | 66.1 ± 0.3 | 66.4 ± 0.3 | 66.5 ± 0.3 | 67.4 ± 0.5 | 67.6 ± 0.1 | **67.9 ± 0.2** | 67.5 ± 0.3 |

Table 11: Performance comparison on ImageNet-1K with decoupled distillation methods evaluated by ResNet-50 and ResNet-101

## D Unified and Fair Framework across Various Methods

To investigate whether our proposed unified RD$^3$ framework introduce potential bias, we provide the performance variations of all methods under this gradual setting in the table below. It can be observed that all methods exhibit similar patterns of performance improvement, which further supports the fairness and consistency of our proposed evaluation protocol. The consistent patterns observed across all methods under massive changeable settings, provide strong evidence for the fairness of our proposed RD$^3$ framework.

| Config | SRe$^2$L | CDA | G-VBSM | DWA | EDC | Minimax | D$^4$M | RDED |
|---|---|---|---|---|---|---|---|---|
| Simplified | 26.6 | 27.2 | 27.1 | 28.7 | 31.3 | 30.7 | 30.5 | 30.9 |
| + Aligned Batchsize | 35.2 | 36.4 | 36.9 | 37.7 | 41.2 | 40.4 | 40.7 | 41.2 |
| + Aligned Data Augmentation | 38.3 | 39.4 | 39.7 | 40.6 | 43.1 | 42.7 | 42.5 | 43.9 |
| + Aligned LRS ($\zeta$) | 40.2 | 41.2 | 41.5 | 42.5 | 46.9 | 45.9 | 45.4 | 46.3 |

Table 12: The performance across various distillation methods with incremental settings. All the methods exhibit the same pattern on performance improvement.

## E More Performance on ResNet Series

To systematically investigate the performance characteristics of synthetic datasets generated by various methods, we conduct comprehensive evaluations using deeper architectures (i.e., ResNet-50 and ResNet-101) that share structural homology with teacher models, assessing performance across multiple compression ratios on ImageNet-1K. Our empirical analysis shown in Table 11 reveals that performance disparities across methods diminish proportionally with model depth escalation, with maximum accuracy variance reduced to merely 2.3% under ResNet-101 evaluation, revealing

that the substantial performance variations reported in existing literature predominantly stem from inconsistent evaluation protocols, and making the efficiency more essential than the effectiveness.

Furthermore, when doubling image quantity to IPC=50 and IPC=100 settings, synthetic datasets demonstrate negligible performance enhancements on ResNet-50/101 compared to ResNet-18 baselines, suggesting current synthesis techniques fail to adequately preserve the original data distribution's topological characteristics. The conspicuous absence of challenging boundary samples and failure in faithful reconstruction of class-discriminative features indicate that current generation-based mechanisms cannot effectively capture distribution extremities. This fundamental limitation in synthesizing distributionally faithful samples, particularly edge-case exemplars, highlights a critical research direction for subsequent investigations in distillation methods.

| Method | ResNet-18 | ResNet-50 | ResNet-101 | MobileNet-V2 | EfficientNet-B0 | Swin-V2-T | ViT-B-16 |
|---|---|---|---|---|---|---|---|
| SRe$^2$L | $55.2 \pm 0.2$ | $57.5 \pm 0.2$ | $59.5 \pm 0.3$ | $31.5 \pm 0.4$ | $44.8 \pm 0.4$ | $55.0 \pm 0.2$ | $51.3 \pm 0.1$ |
| CDA | $56.7 \pm 0.6$ | $58.8 \pm 0.3$ | $60.1 \pm 0.3$ | $33.6 \pm 0.4$ | $46.7 \pm 0.3$ | $57.1 \pm 0.5$ | $52.2 \pm 0.2$ |
| G-VBSM | $56.6 \pm 0.2$ | $58.2 \pm 0.4$ | $60.2 \pm 0.4$ | $33.0 \pm 0.2$ | $47.2 \pm 0.3$ | $58.6 \pm 0.3$ | $56.6 \pm 0.4$ |
| DWA | $57.7 \pm 0.5$ | $59.4 \pm 0.1$ | $61.2 \pm 0.2$ | $30.2 \pm 0.4$ | $47.7 \pm 0.2$ | $58.9 \pm 0.5$ | $54.7 \pm 0.5$ |
| EDC | $60.1 \pm 0.3$ | $62.2 \pm 0.2$ | $62.3 \pm 0.3$ | $38.9 \pm 0.1$ | $50.5 \pm 0.2$ | $62.0 \pm 0.4$ | $59.9 \pm 0.3$ |
| Minimax | $\mathbf{60.4 \pm 0.2}$ | $62.2 \pm 0.3$ | $61.6 \pm 0.3$ | $37.8 \pm 0.4$ | $51.6 \pm 0.1$ | $61.9 \pm 0.2$ | $61.8 \pm 0.2$ |
| D$^4$M | $60.2 \pm 0.4$ | $63.1 \pm 0.2$ | $\mathbf{62.5 \pm 0.3}$ | $\mathbf{39.9 \pm 0.3}$ | $\mathbf{52.0 \pm 0.1}$ | $\mathbf{63.0 \pm 0.3}$ | $\mathbf{62.6 \pm 0.4}$ |
| RDED | $58.9 \pm 0.7$ | $\mathbf{62.3 \pm 0.2}$ | $61.8 \pm 0.4$ | $39.2 \pm 0.3$ | $50.0 \pm 0.3$ | $61.8 \pm 0.2$ | $60.7 \pm 0.3$ |

Table 13: Generalization ability of synthetic dataset on ImageNet-1K under IPC=50 with 50 batch size in post-evaluation phase. The performance degradation is obvious on certain model architectures.

## F    VARYING GENERALIZATION ABILITY

We conducted two supplementary experiments to further investigate the generalization capabilities of synthetic datasets. First, under IPC=50 setting, we adjusted BS from 100 to 50. Experimental results shown in Table 13 reveal significant architectural disparities in BS sensitivity: CNN-based models exhibited 3%-4% performance degradation on ResNet-50/101, while MobileNet-V2 and EfficientNet-B0 architectures suffered over 15% performance drop, indicating substantial variance in gradient fluctuation tolerance across architectures, particularly in data-efficient learning scenarios like dataset distillation. Conversely, ViT-based models demonstrated remarkable stability with merely 1% degradation on ViT-B-16 and even performance improvement on Swin-V2-T variants, confirming ViT's training stability given fixed dataset settings.

Subsequently, we evaluated cross-architecture generalization under IPC=10 as shown in Table 14. For CNN-based models, performance degradation remained acceptable compared to IPC=50 baselines, showing comparable decline patterns to ResNet-18 observations. However, ViT-based architectures suffered catastrophic performance collapse, with both Swin-V2-T and ViT-B-16 variants experiencing over 40% accuracy reduction. This phenomenon aligns with established observations regarding Vision Transformers' limited efficacy in low-sample regimes, simultaneously presenting critical challenges for achieving successful knowledge transfer from CNN-optimized distilled datasets to ViT architectures under high compression ratios.

| Method | ResNet-18 | ResNet-50 | ResNet-101 | MobileNet-V2 | EfficientNet-B0 | Swin-V2-T | ViT-B-16 |
|---|---|---|---|---|---|---|---|
| SRe$^2$L | $40.2 \pm 0.3$ | $48.5 \pm 0.3$ | $45.1 \pm 0.2$ | $33.0 \pm 0.3$ | $43.3 \pm 0.5$ | $15.5 \pm 0.2$ | $11.2 \pm 0.2$ |
| CDA | $41.2 \pm 0.6$ | $49.2 \pm 0.3$ | $46.6 \pm 0.3$ | $33.4 \pm 0.4$ | $42.7 \pm 0.4$ | $16.3 \pm 0.2$ | $10.2 \pm 0.2$ |
| G-VBSM | $41.5 \pm 0.6$ | $49.5 \pm 0.3$ | $46.4 \pm 0.2$ | $34.5 \pm 0.4$ | $43.8 \pm 0.4$ | $19.4 \pm 0.3$ | $11.8 \pm 0.3$ |
| DWA | $42.5 \pm 0.7$ | $50.1 \pm 0.3$ | $48.7 \pm 0.1$ | $36.5 \pm 0.5$ | $45.4 \pm 0.1$ | $18.8 \pm 0.2$ | $13.6 \pm 0.2$ |
| EDC | $46.9 \pm 0.6$ | $53.9 \pm 0.3$ | $52.8 \pm 0.4$ | $39.8 \pm 0.2$ | $48.4 \pm 0.3$ | $27.7 \pm 0.5$ | $22.1 \pm 0.2$ |
| Minimax | $45.9 \pm 0.7$ | $\mathbf{54.7 \pm 0.2}$ | $52.4 \pm 0.4$ | $38.1 \pm 0.5$ | $\mathbf{49.6 \pm 0.2}$ | $\mathbf{28.4 \pm 0.1}$ | $\mathbf{23.1 \pm 0.3}$ |
| D$^4$M | $45.4 \pm 0.6$ | $53.3 \pm 0.2$ | $53.5 \pm 0.2$ | $39.8 \pm 0.4$ | $47.9 \pm 0.3$ | $22.6 \pm 0.1$ | $22.1 \pm 0.2$ |
| RDED | $\mathbf{46.3 \pm 0.2}$ | $53.2 \pm 0.3$ | $\mathbf{53.7 \pm 0.2}$ | $\mathbf{40.2 \pm 0.4}$ | $48.2 \pm 0.4$ | $28.1 \pm 0.3$ | $22.8 \pm 0.1$ |

Table 14: Generalization ability of synthetic dataset on ImageNet-1K under IPC=10. The ViT-based models show extremely low performance with high compression ratio.

| | | ResNet-18 | | | | | | | |
|---|---|---|---|---|---|---|---|---|---|
| Dataset | IPC | Optimization | | | | | Generation | | Selection |
| | | SRe$^2$L | CDA | G-VBSM | DWA | EDC | Minimax | D$^4$M | RDED |
| CIFAR10 | 1 | 10.5 ±0.5 | 10.3 ±0.4 | 9.7 ±0.6 | 9.2 ±0.5 | **18.8 ±0.7** | - | 17.1 ±0.5 | 12.2 ±0.6 |
| | 10 | 14.1 ±0.4 | 14.6 ±0.3 | 16.6 ± 0.3 | 18.1 ±0.2 | 23.1 ±0.5 | - | 24.2 ±0.4 | 22.8 ±0.3 |
| | 50 | 15.6 ±0.4 | 13.7 ±0.4 | 17.2 ±0.5 | 22.2 ±0.5 | 29.2 ±0.4 | - | 30.8 ±0.5 | 35.3 ±0.3 |
| | 100 | 18.2 ±0.4 | 18.5 ±0.4 | 20.3 ±0.3 | 27.5 ±0.2 | 39.4 ±0.3 | - | 38.5 ±0.3 | 41.6 ±0.5 |
| CIFAR100 | 1 | 1.8 ±0.4 | 1.6 ±0.4 | 1.6 ±0.3 | 2.1 ±0.5 | 3.7 ±0.4 | - | 4.3 ±0.4 | **4.4 ±0.7** |
| | 10 | 3.2 ±0.3 | 3.0 ±0.6 | 4.1 ±0.6 | 3.8 ±0.7 | **12.4 ±0.5** | - | 8.7 ±0.6 | 11.6 ±0.4 |
| | 50 | 4.9 ±0.4 | 5.4 ±0.3 | 5.0 ±0.5 | 5.9 ±0.4 | 21.4 ±0.5 | - | 15.1 ±0.3 | 23.6 ±0.5 |
| | 100 | 7.5 ±0.3 | 7.4 ±0.4 | 7.8 ±0.3 | 8.9 ±0.5 | 30.2 ±0.6 | - | 28.7 ±0.4 | 32.5 ±0.3 |
| TinyImageNet | 1 | 0.9 ±0.5 | 1.0 ±0.4 | 1.3 ±0.6 | 1.9 ±0.5 | **3.3 ±0.7** | 2.2 ±0.5 | 2.1 ±0.6 | 3.2 ±0.5 |
| | 10 | 1.9 ±0.5 | 2.2 ±0.4 | 2.9 ±0.4 | 4.3 ±0.6 | 9.7 ±0.5 | 6.3 ±0.5 | 4.7 ±0.4 | 10.6 ±0.5 |
| | 50 | 5.3 ±0.4 | 7.3 ±0.3 | 7.2 ±0.5 | 11.7 ±0.5 | 20.3 ±0.5 | 18.4 ±0.3 | 8.9 ±0.5 | 22.8 ±0.7 |
| | 100 | 10.1 ±0.3 | 12.7 ±0.4 | 13.3 ±0.4 | 15.1 ±0.2 | 27.8 ±0.4 | 25.3 ±0.4 | 12.1 ±0.6 | 30.7 ±0.3 |
| ImageNette | 1 | 18.2 ±0.4 | 18.7 ±0.5 | 18.3 ±0.5 | 16.3 ±0.7 | **26.7 ±0.6** | 18.9 ±0.4 | 22.4 ±0.3 | 22.5 ±0.3 |
| | 10 | 20.2 ±0.4 | 21.5 ±0.5 | 21.2 ±0.5 | 29.3 ±0.3 | 38.6 ±0.7 | 39.1 ±0.4 | **40.4 ±0.3** | 34.6 ±0.6 |
| | 50 | 25.4 ±0.1 | 27.8 ±0.4 | 28.3 ±0.5 | 32.5 ±0.5 | 46.4 ±0.4 | 57.6 ±0.5 | **61.6 ±0.7** | 50.7 ±0.64 |
| | 100 | 30.0 ±0.4 | 31.5 ±0.3 | 31.2 ±0.4 | 37.3 ±0.5 | 52.7 ±0.5 | **68.5 ±0.6** | 66.4 ±0.3 | 59.4 ±0.5 |
| ImageWoof | 1 | 11.7 ±0.6 | 12.2 ±0.8 | 11.3 ±0.6 | 12.5 ±0.7 | 12.6 ±0.5 | 16.8 ±0.4 | 13.6 ±0.5 | **19.0 ±0.7** |
| | 10 | 14.4 ±0.4 | 12.4 ±0.3 | 13.2 ±0.5 | 16.8 ±0.6 | 23.7 ±0.4 | **24.3 ±0.2** | 22.4 ±0.4 | 21.1 ±0.5 |
| | 50 | 15.6 ±0.4 | 13.6 ±0.6 | 14.3 ±0.6 | 23.7 ±0.4 | 25.8 ±0.5 | **41.2 ±0.5** | 31.4 ±0.3 | 31.2 ±0.4 |
| | 100 | 18.5 ±0.5 | 19.1 ±0.6 | 17.8 ±0.3 | 28.3 ±0.2 | 30.4 ±0.4 | **49.7 ±0.3** | 42.2 ±0.4 | 42.9 ±0.3 |
| ImageNet-1K | 1 | 0.3 ±0.2 | 0.2 ± 0.6 | 0.2 ± 0.3 | 0.4 ± 0.5 | 0.6 ± 0.3 | **1.3 ± 0.7** | 0.7 ± 0.5 | 1.1 ± 0.4 |
| | 10 | 1.4 ± 0.6 | 1.5 ± 0.4 | 1.2 ± 0.3 | 1.9 ± 0.3 | 6.2 ± 0.4 | 9.2 ± 0.3 | 5.6 ± 0.2 | **12.4 ± 0.4** |
| | 50 | 3.5 ± 0.3 | 5.9 ± 0.4 | 7.2 ± 0.5 | 6.1 ± 0.4 | 17.4 ± 0.5 | **33.4 ± 0.4** | 18.9 ± 0.5 | 31.7 ± 0.4 |
| | 100 | 4.6 ± 0.2 | 7.3 ± 0.1 | 15.2 ± 0.2 | 16.9 ± 0.4 | 21.1 ± 0.3 | **42.1 ± 0.1** | 26.1 ± 0.2 | 40.1 ± 0.4 |

Table 15: Performance comparison across various datasets with well-known decoupled distillation methods using hard label. All the methods exhibit significant performance degradation.

# G  HARD LABEL PERFORMANCE

To systematically investigate the role of soft labels in decoupled dataset distillation, we conducted experiments replacing soft labels with one-hot labels during evaluation while keeping other settings unchanged.

Experimental results shown in Table 15 reveal that optimization-based methods exhibit intolerable performance degradation across all datasets. Due to their exclusive reliance on single teacher models during optimization, the generated images tend to overfit to specific parameters. Simultaneously, using only cross-entropy loss and matching BN statistics fails to effectively help randomly initialized student models learn meaningful categorical information. This forces optimization-based methods to completely depend on teacher-generated soft labels for knowledge transfer, creating significant deployment challenges for lightweight and simplified distillation implementations.

For generation-based methods, while Minimax remains inapplicable to ImageNet and external subsets, its integration with pretrained DiT models enables generation of near-photorealistic images preserving substantial category-related features on ImageWoof, ImageNette, and ImageNet-1K datasets. This facilitates effective learning of mapping relationships between generated images and their labels in student models. However, D$^4$M's complete dependence on latent distributions in Stable Diffusion results in significant divergence from target dataset distributions, especially on TinyImageNet. Without soft label guidance, D$^4$M's excessive diversity hinders accurate student learning. Both generation-based methods perform poorly on TinyImageNet datasets, indicating resolution differences exacerbate distributional inconsistencies.

Selection-based method demonstrate superior performance across most datasets by preserving authentic category-related visual features through real image selection. On fine-grained datasets like ImageWoof, RDED enhances dataset representativeness through strategic simple image selection. However, on coarse-grained datasets like ImageNette, oversimplified images impair student learning, resulting in substantial performance gaps compared to generation-based methods. These findings suggest future improvements should focus on developing automated mechanisms to identify dataset

distribution characteristics and impose corresponding constraints, potentially enabling universal algorithms adaptable to various dataset types.

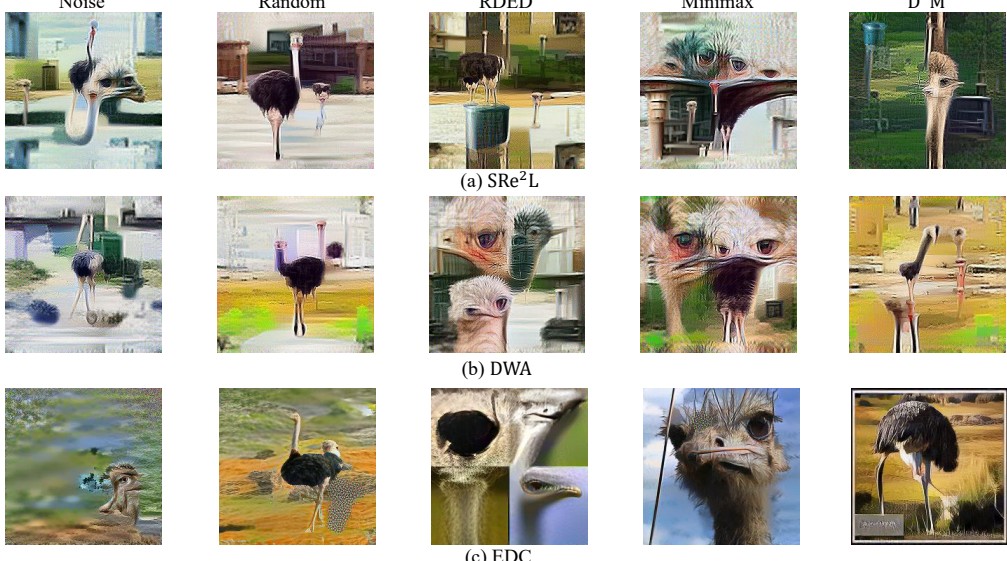

Figure 5: Visual comparison of class "ostrich" with different distillation methods using various initialization.

## H    VISUAL COMPARISON OF INITIALIZATION

To visually demonstrate the impact of different initializations on optimization-based methods, we present the corresponding visualizations in Figure 5. It is evident that varying initializations substantially influence the final synthesized images.

For SRe$^2$L, limited visual divergence across initializations arises because its optimization process aligns closely with cross-entropy loss and BN statistics, prioritizing distributional alignment over diversity. The modest performance improvement primarily stems from generated images better matching the data distribution learned by the teacher model.

For DWA, initializing with real images imposes distributional constraints during optimization. While this introduces perturbations to the teacher model and disentangles BN statistics to enhance diversity, it also causes performance degradation when using noise or distribution-shifted D$^4$M images, as the teacher model struggles to transfer knowledge accurately under such conditions.

For EDC, the collaboration of multiple teacher models and fewer optimization iterations leads to severe performance deterioration when noise-based initializations are employed. As shown in the figure, images generated by EDC in this scenario resemble noise. However, when initialized with more representative samples, EDC's diversity becomes constrained. With fewer optimization steps, the generated images closely resemble the initialization, preserving category-relevant details and consequently improving performance.

With the qualitative and quantitative analysis above, We identify the utilization of more powerful initialization should not be considered as a strong contribution.

## I    HYBRID LABEL EXTENSION

As shown in Table 16, we further investigate the impact of hybrid soft labels across additional datasets and compression ratios. To align with the setting of EDC, we employ ResNet-18, ConvNet-W-128, WideResNet-16-2, MobileNet-V2, and ShuffleNet-V2-X0-5 to generate hybrid soft labels for TinyImageNet, while ResNet-18, MobileNet-v2, ShuffleNet-V2-X0-5, and AlexNet is utilized to produce hybrid soft labels for ImageNet-1K. It is observable that, regardless of the target dataset's scale, hybrid soft labels yield substantial performance improvements under IPC=1 and IPC=10.

| IPC | Hybrid | SRe$^2$L | CDA | G-VBSM | DWA | EDC | Minimax | D$^4$M | RDED |
|-----|--------|----------|-----|--------|-----|-----|---------|--------|------|
| | | | | | TinyImageNet | | | | |
| 1 | - | 6.1 | 7.1 | 6.2 | 6.8 | 10.2 | 9.8 | 3.9 | **11.1** |
| | ✓ | 13.8 (7.7 ↑) | 14.6 (7.5 ↑) | 14.7 (8.5 ↑) | 15.3 (8.5 ↑) | **19.5 (9.3 ↑)** | 18.2 (8.4 ↑) | 12.5 (8.6 ↑) | 19.1 (8.0 ↑) |
| 10 | - | 34.2 | 37.5 | 37.3 | 38.3 | 42.1 | 39.4 | 36.7 | **44.2** |
| | ✓ | 38.7 (4.5 ↑) | 42.2 (4.7 ↑) | 42.4 (5.1 ↑) | 43.6 (5.3 ↑) | 48.0 (5.9 ↑) | 44.6 (5.2 ↑) | 41.1 (4.4 ↑) | **49.2 (5.0 ↑)** |
| 50 | - | 52.5 | 53.0 | 53.7 | 54.2 | 57.1 | 54.4 | 53.8 | **58.7** |
| | ✓ | 48.9 (3.6 ↓) | 48.7 (4.3 ↓) | 49.3 (4.4 ↓) | 48.8 (5.4 ↓) | 52.3 (4.8 ↓) | 50.1 (4.3 ↓) | 49.2 (4.6 ↓) | 53.1 (5.6 ↓) |
| | | | | | ImageNet-1K | | | | |
| 1 | - | 4.1 | 4.2 | 4.2 | 4.5 | 7.0 | 6.8 | 5.4 | **7.6** |
| | ✓ | 12.2 (8.1 ↑) | 12.5 (8.3 ↑) | 12.8 (8.6 ↑) | 13.6 (9.1 ↑) | 15.5 (8.5 ↑) | **15.7 (8.9 ↑)** | 13.9 (8.5 ↑) | 15.6 (8.0 ↑) |
| 10 | - | 40.2 | 41.2 | 41.5 | 42.5 | **46.9** | 45.9 | 45.4 | 46.3 |
| | ✓ | 40.9 (0.7 ↑) | 42.1 (0.9 ↑) | 42.3 (0.8 ↑) | 43.7 (1.2 ↑) | **47.9 (1.0 ↑)** | 46.8 (0.9 ↑) | 46.1 (0.7 ↑) | 47.5 (1.2 ↑) |
| 50 | - | 55.2 | 56.7 | 56.6 | 57.7 | 60.1 | **60.4** | 60.2 | 58.9 |
| | ✓ | 51.2 (4.0 ↓) | 54.3 (2.4 ↓) | 52.4 (4.2 ↓) | 54.9 (2.8 ↓) | **57.1 (3.0 ↓)** | 56.8 (3.6 ↓) | 56.3 (3.9 ↓) | 56.4 (2.5 ↓) |

Table 16: performance of using hybrid label on TinyImageNet and ImageNet-1k. The performance gain decrease with the growing IPC.

Notably, even for methods devoid of proxy model involvement, such as Minimax and D$^4$M, hybrid soft labels consistently enhance performance. We hypothesize that at lower IPC levels, diverse teacher models effectively augment dataset diversity through soft labels, thereby improving performance despite architectural discrepancies with the student model.

However, under IPC=50, hybrid soft labels induce significant performance degradation. We attribute this phenomenon to the fact that the generated images already ensure sufficient diversity, whereas overly heterogeneous soft labels hinder the student model's ability to learn precise categorical information from the distilled dataset, leading to performance decline. Synthesizing these observations, we emphasize that for dataset distillation tasks, the optimal selection of soft label formulations must be adaptively tailored to specific settings. Furthermore, exploring superior strategies for teacher model ensemble design across different methods remains a critical direction for future research.

| IPC | Loss | SRe$^2$L | CDA | G-VBSM | DWA | EDC | Minimax | D$^4$M | RDED |
|-----|------|----------|-----|--------|-----|-----|---------|--------|------|
| 1 | KL | 4.1 | 4.2 | 4.2 | 4.5 | 7.0 | 6.8 | 5.4 | **7.6** |
| | GT | 0.3 (3.8 ↓) | 0.2 (4.0 ↓) | 0.2 (4.0 ↓) | 0.4 (4.1 ↓) | 0.6 (6.4 ↓) | **1.3 (5.5 ↓)** | 0.7 (4.7 ↓) | 1.1 (6.5 ↓) |
| | MSE-GT | 3.6 (0.5 ↓) | 4.6 (0.4 ↑) | 4.8 (0.6 ↑) | 5.2 (0.7 ↑) | 6.8 (0.2 ↓) | 7.5 (0.7 ↑) | 5.7 (0.3 ↑) | **7.8 (0.2 ↑)** |
| 10 | KL | 40.2 | 41.2 | 41.5 | 42.5 | **46.9** | 45.9 | 45.4 | 46.3 |
| | GT | 1.4 (38.8 ↓) | 1.5 (39.7 ↓) | 1.2 (40.3 ↓) | 1.9 (40.6 ↓) | 6.2 (40.7 ↓) | 9.2 (36.7 ↓) | 5.6 (39.8 ↓) | **12.4 (33.9 ↓)** |
| | MSE-GT | 40.9 (0.7 ↑) | 42.0 (0.8 ↑) | 42.3 (0.8 ↑) | 43.1 (0.6 ↑) | **47.9 (1.0 ↑)** | 47.2 (1.3 ↑) | 47.5 (2.1 ↑) | 46.8 (0.5 ↑) |
| 50 | KL | 55.2 | 56.7 | 56.6 | 57.7 | 60.1 | **60.4** | 60.2 | 58.9 |
| | GT | 3.5 (51.7 ↓) | 5.9 (50.8 ↓) | 7.2 (49.4 ↓) | 6.1 (51.6 ↓) | 17.4 (42.7 ↓) | 33.4 (27.0 ↓) | 18.9 (41.3 ↓) | 31.7 (27.2 ↓) |
| | MSE-GT | 56.4 (1.2 ↑) | 58.2 (1.5 ↑) | 57.8 (1.2 ↑) | 59.1 (1.4 ↑) | 60.8 (0.7 ↑) | **61.5 (1.1 ↑)** | 61.3 (1.1 ↑) | 60.2 (1.3 ↑) |
| 100 | KL | 59.7 | 60.6 | 61.5 | 62.1 | 63.2 | 62.2 | **63.5** | 61.5 |
| | GT | 4.6 (55.1 ↓) | 7.3 (53.3 ↓) | 15.2 (46.3 ↓) | 16.9 (45.2 ↓) | 21.1 (42.1 ↓) | **42.1 (20.1 ↓)** | 26.1 (37.4 ↓) | 40.1 (21.4 ↓) |
| | MSE-GT | 59.5 (0.2 ↓) | 60.1 (0.5 ↓) | 61.9 (0.4 ↑) | 62.0 (0.1 ↓) | **64.1 (0.9 ↑)** | 63.0 (0.8 ↑) | 62.9 (0.6 ↓) | 62.7 (1.2 ↑) |

Table 17: performance of using different loss functions on ImageNet-1K. The performance could be further enhanced with the appropriate loss fuction.

## J   LOSS FUNCTION CONSIDERATION

As shown in Table 17, we evaluate the performance of student models under different loss functions across all methods, with experimental results presented in the table. When using only the cross-entropy loss with ground-truth labels (equivalent to the hard label paradigm), all methods exhibit significant performance degradation, as detailed in our analysis of hard label performance. In contrast, when combining the cross-entropy losses between student outputs and both teacher outputs and hard labels as a joint loss function, effective performance improvements are achieved across most settings. For experimental simplicity, we did not extensively tune the weights of the two cross-entropy losses, suggesting that optimized parameter settings could yield further enhancements. Future work should

focus on designing more effective loss functions tailored to the characteristics of the distilled dataset, thereby facilitating improved knowledge acquisition by student models.

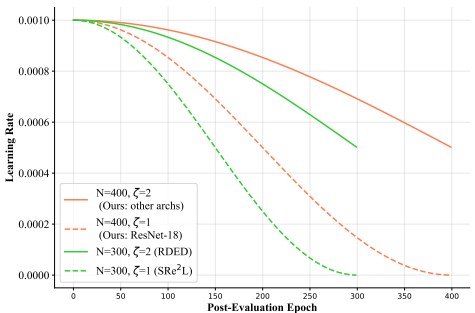

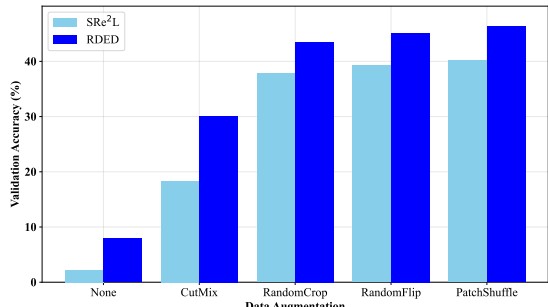

Figure 6: Comparison of the learning rate decay with different post-evaluation epoch and smoothing factor. Our PD$^3$ framework provide a more refined parameter selection.

Figure 7: Comparison of the impact of using different data augmentation on ImageNet-1K under IPC=10. CutMix and RandomCrop play an essential role in enhancing performance.

## K LR SCHEDULER ANALYSIS

The visual comparative analysis of learning rate decay strategies employed by RD$^3$ across different settings and those used in prior works is illustrated in the figure. Methods like SRe$^2$L and CDA adopt a cosine decay strategy with $\zeta=1$, while RDED and EDC propose that using $\zeta=2$ further enhances performance. Recent work CVDD suggests adapting $\zeta$ based on dataset compression ratios and evaluation models. Under extended training epochs, we implement refined $\zeta$ selection according to evaluation models and visualized it as shown in Figure 6. for ResNet, $\zeta=1$ is chosen, positioning the learning rate curve between historical settings. This ensures faster decay without premature optimization termination, thereby achieving additional performance gains. For other evaluation models, we employ larger learning rates than all previous methods under equivalent training epochs. Consequently, student models learn with larger step sizes despite imperfect knowledge alignment, enabling escape from local optima while accelerating convergence. To maintain framework simplicity, we did not exhaustively optimize $\zeta$ selection, suggesting that adjusting $\zeta$ in specific scenarios could potentially achieve superior performance.

## L SOFT LABEL TEMPERATURE

Under the unified setting provided by RD$^3$, we investigate whether the soft label temperature differentially impacts performance based on variations in dataset generation processes and data distributions. Experimental results, illustrated in Figure 8, reveal consistent trends across three representative methods. At excessively low temperatures, all methods exhibit pronounced performance degradation, attributed to the over-concentrated output distribution of the teacher model, which resembles hard labels and fails to provide nuanced prior knowledge for the student model. When temperatures exceed 20, performance plateaus or even declines in certain methods, as overly smoothed soft labels from

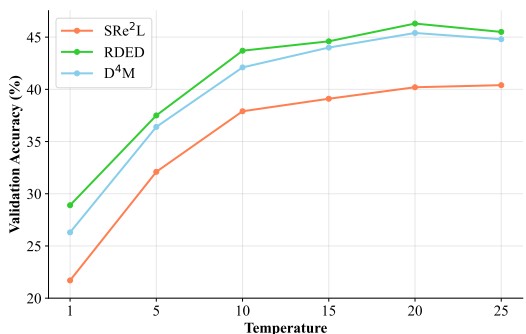

Figure 8: The performance of ResNet-18 trained on different temperature settings on ImageNet-1K under IPC=10.

the teacher model obscure categorical discriminability. These observations align with phenomena identified in our experiments with different loss functions, further substantiating that under the soft label paradigm, variations in generated images do not fundamentally alter behavioral characteristics, with soft labels predominantly governing the efficacy of knowledge transfer.

## M    IMPACT OF DATA AUGMENTATIONS

To investigate the generalizability of data augmentation, we designed ablation studies on various methods, with results illustrated in Figure 7. Without any data augmentation, all decoupled dataset distillation methods exhibit extremely poor performance. With CutMix and RandomResizedCrop added, both augmentation strategies substantially enhance the performance of all methods. Cropping images and compositing patches from different images significantly improve dataset diversity, enabling teacher models to convey richer knowledge via soft labels. Further deploying RandomHorizontalFlip yields an additional around 1% improvement, demonstrating that simple flipping operations still contribute meaningfully. Finally, we tested the generalization of the PatchShuffle strategy proposed in RDED. While PatchShuffle randomly replaces patches across images, specifically designed for RDED's patch-composed images, surprisingly, applying PatchShuffle to SRe$^2$L whose optimization process is patch-agnostic still achieves 1% performance gains. This confirms that diverse augmentation operations universally enhance dataset diversity and boost performance. Consequently, future work should explicitly disclose whether additional data augmentations are employed and include corresponding ablation analyses.

## N    IMPACT OF DIVERSE SOFT LABEL ENHANCEMENT TECHNIQUES

In the context of decoupled dataset distillation, the use of soft labels serves as the foundation for applying data augmentation (e.g., CutMix) during the post-evaluation phase. Therefore, our intention is to highlight that any absolute performance gain claimed by newly proposed methods over existing baselines must be evaluated under identical post-evaluation settings, including the use of soft labels and data augmentation strategies. To investigate the impact of different knowledge distillation techniques to the dataset distillation, we have incorporated additional soft label augmentation techniques during the post-evaluation phase. The experimental results on ImageNet-1K under IPC=10 are shown in Table 18. As observed, soft label augmentation can act as a general performance booster. However, when applied without proper constraints, it may lead to unfair comparisons. This observation further exposes the problem we have identified in the current decoupled dataset distillation literature and underscores the necessity of our proposed work.

| Config | SRe$^2$L | CDA | G-VBSM | DWA | EDC | Minimax | D$^4$M | RDED |
|---|---|---|---|---|---|---|---|---|
| None | 40.2 | 41.2 | 41.5 | 42.5 | 46.9 | 45.9 | 45.4 | 46.3 |
| DKD (Zhao et al., 2022) | 41.3 | 42.4 | 42.2 | 43.3 | 47.8 | 47.1 | 46.5 | 47.4 |
| NKD (Yang et al., 2023) | 41.1 | 42.7 | **42.6** | 43.0 | 48.5 | 47.9 | **46.9** | 48.0 |
| LSKD (Sun et al., 2024b) | 41.5 | 43.0 | 42.2 | 43.4 | 48.7 | 48.4 | 46.8 | **48.4** |
| CRLD (Zhang et al., 2024) | **41.8** | **43.2** | 42.5 | **43.7** | **45.1** | **48.8** | 46.2 | 48.2 |

Table 18: Impact of soft label enhancement techniques. It is clear to see that using the more powerful knowledge distillation methods could lead to a consistent and significant performance improvement across various distilled datasets.

## O    QUALITATIVE INTERPRETATION

### O.1    TRAINING DYNAMIC ANALYSIS

Under our proposed framework, we observe significant performance differences in generated datasets from various methods during student model training. The training accuracy and test accuracy curves of student models during evaluation are shown in Figure 9. Using randomly sampled images as the baseline, we note that the real training process achieves high generalization due to extensive data augmentation, reflected in the large gap between the two curves.

For optimization-based methods, since the generated datasets are optimized through cross-entropy loss and global BN statistics, the images tend to be overly simplistic for student models. This results in high training accuracy but low test accuracy. The gap between training and test accuracy gradually narrows as method performance improves. The only exception is G-VBSM, whose generated datasets exhibit increased complexity but lack explicit guidance due to the introduction of auxiliary models during optimization.

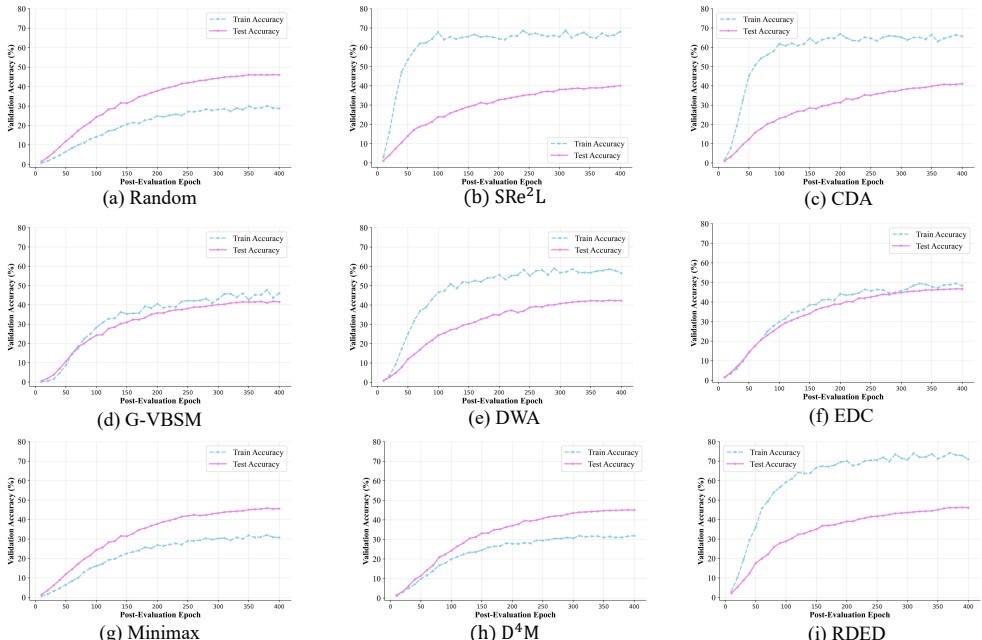

Figure 9: Training dynamics comparison between various training datasets on ImageNet under IPC=10. Different methods exhibit significant difference of gap between training and test accuracy.

Generation-based methods demonstrate training dynamics nearly identical to real datasets, confirming that diffusion models can effectively approximate real data distributions while preserving categorical information and generating diverse images. Future work should explore how to identify and produce more beneficial data distributions based on this foundation.

In contrast, selection-based method prioritize images based on classifier accuracy, achieving the highest training accuracy. Despite this, RDED shows competitive performance under high compression ratios, surpassing all methods except EDC. However, under lower compression settings, RDED's performance declines sharply due to insufficient dataset diversity, highlighting a critical direction for future optimization.

## O.2 T-SNE VISUAL ANALYSIS

We explore the differences in synthetic datasets produced by various distillation methods from another perspective. By visualizing the feature distributions of generated datasets and the original dataset using t-SNE, as shown in Figure 10, we can intuitively observe variations in data distributions across methods.

For optimization-based methods, except for EDC, whose data distribution aligns closely with the original dataset, other methods exhibit significant distribution shifts. In fine-grained categories, images generated by SRe$^2$L, CDA, and DWA become overly simplistic, preventing teacher models from providing effective guidance. while for G-VBSM, although its generated dataset demonstrates dispersed intra-class distributions, inter-class distances are inappropriately minimized in incorrect directions.

Generation-based approaches (i.e., D$^4$M and Minimax) show data distributions largely consistent with the original dataset, confirming that diffusion models effectively approximate real data distributions, thereby achieving competitive performance under low compression ratios.

The selection-based method RDED, which selects images based on classifier accuracy, generates data distributions similar to optimization-based methods. However, since its dataset still comprises original images, it maintains favorable intra-class diversity. Nevertheless, under low compression ratios, RDED also faces challenges of distribution shifts.

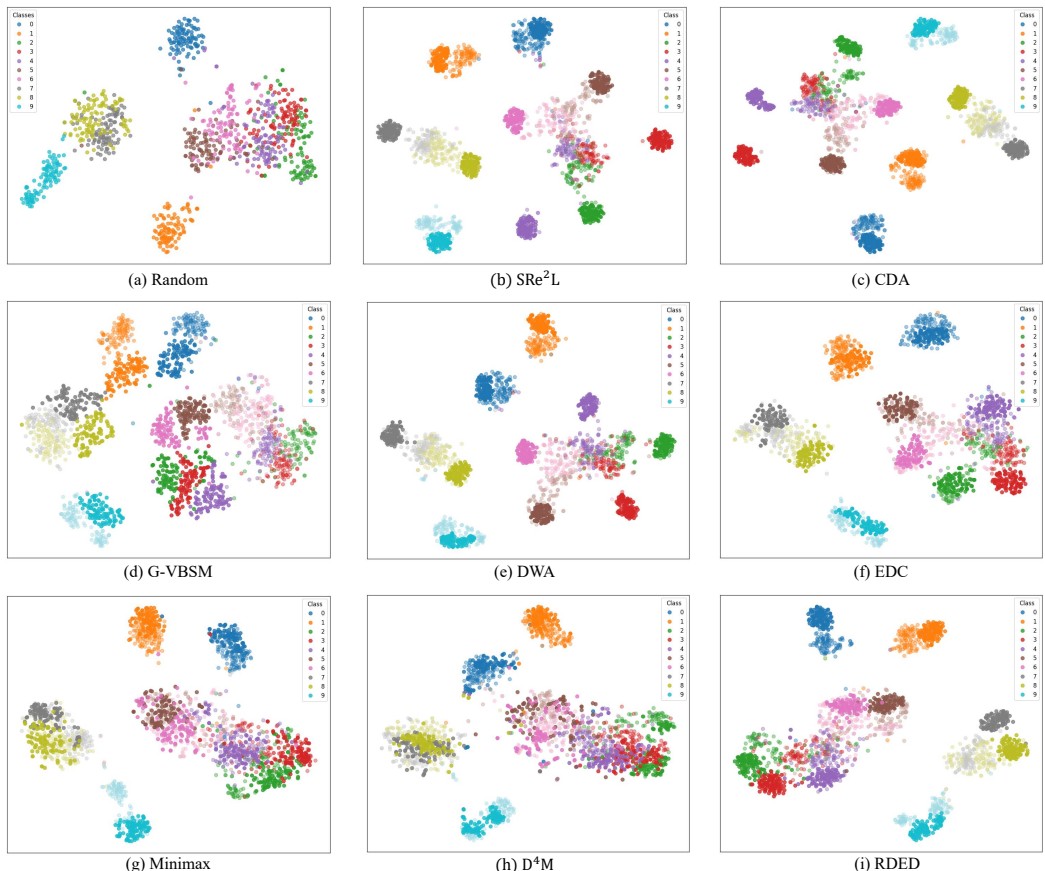

Figure 10: T-SNE visualizations of top 10 classes in ImageNet-1K from different synthetic datasets under IPC=100. The dark dots and light dots denote the synthetic datasets and real dataset respectively. For a clear comparison, we additionally provide the distribution of only real dataset shown in (a).

## P    VISUALIZATION

We provide visualizations of the images sampled from real dataset and synthetic datasets from different methods, as illustrated Figures 11 to 19.

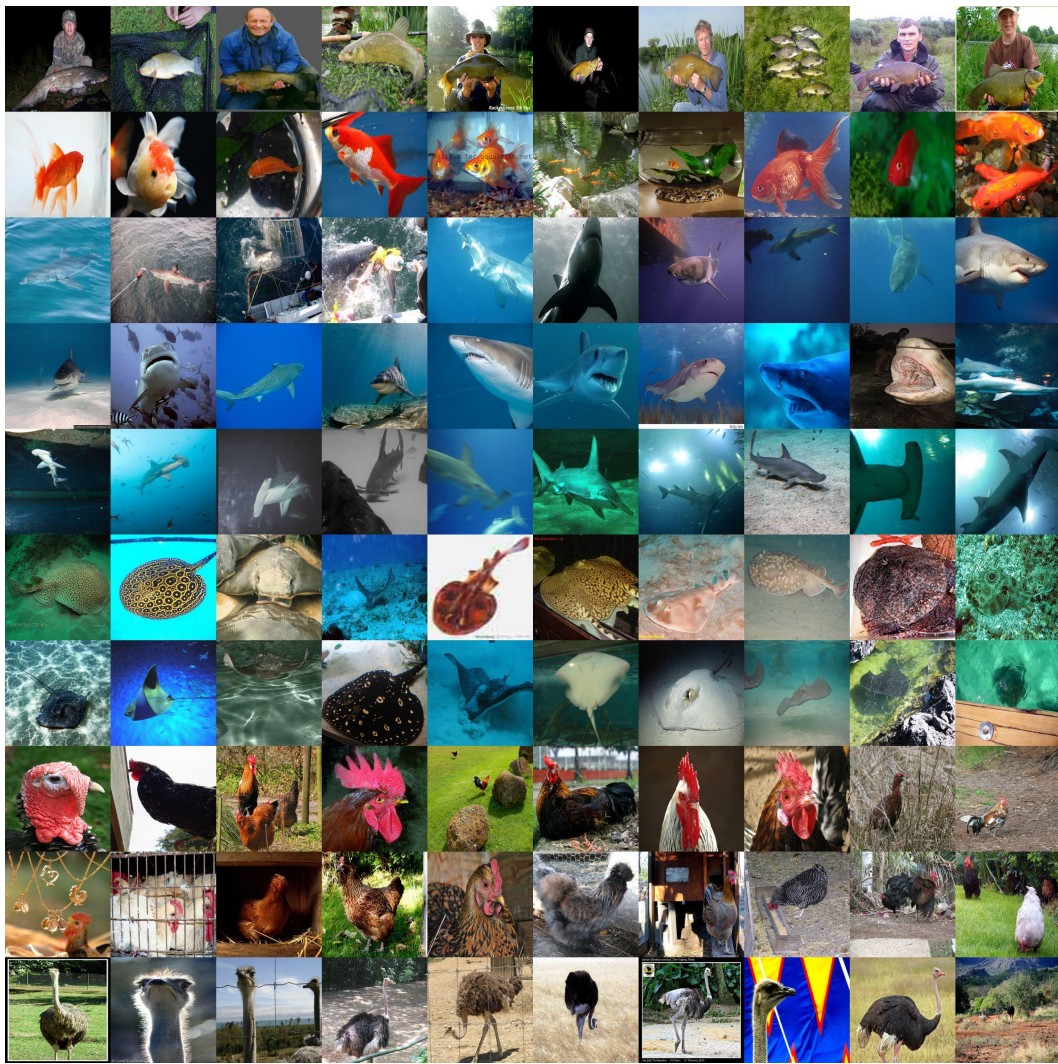

Figure 11: Visualization of top 10 classes in ImageNet-1K from real dataset.

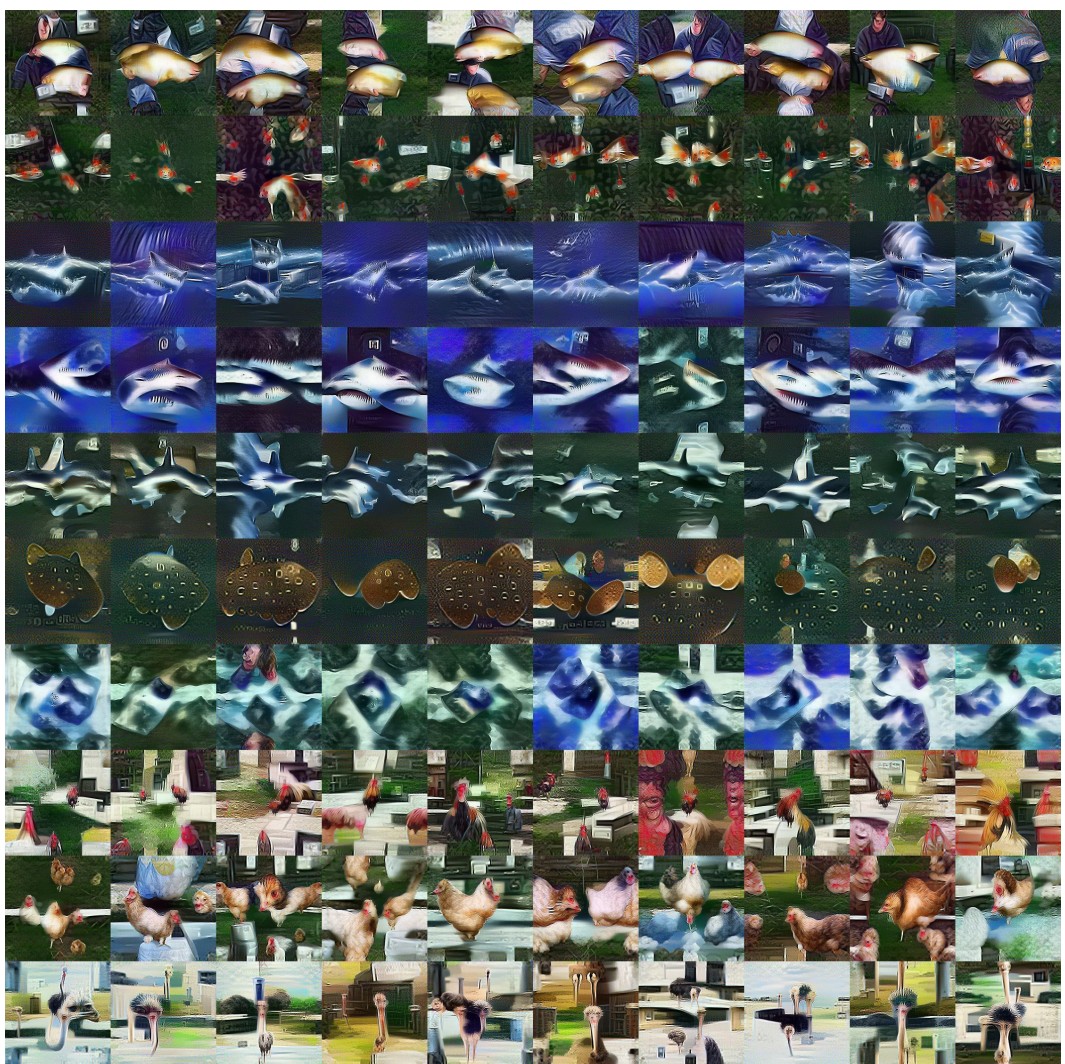

Figure 12: Visualization of top 10 classes in ImageNet-1K from SRe$^2$L under IPC=100.

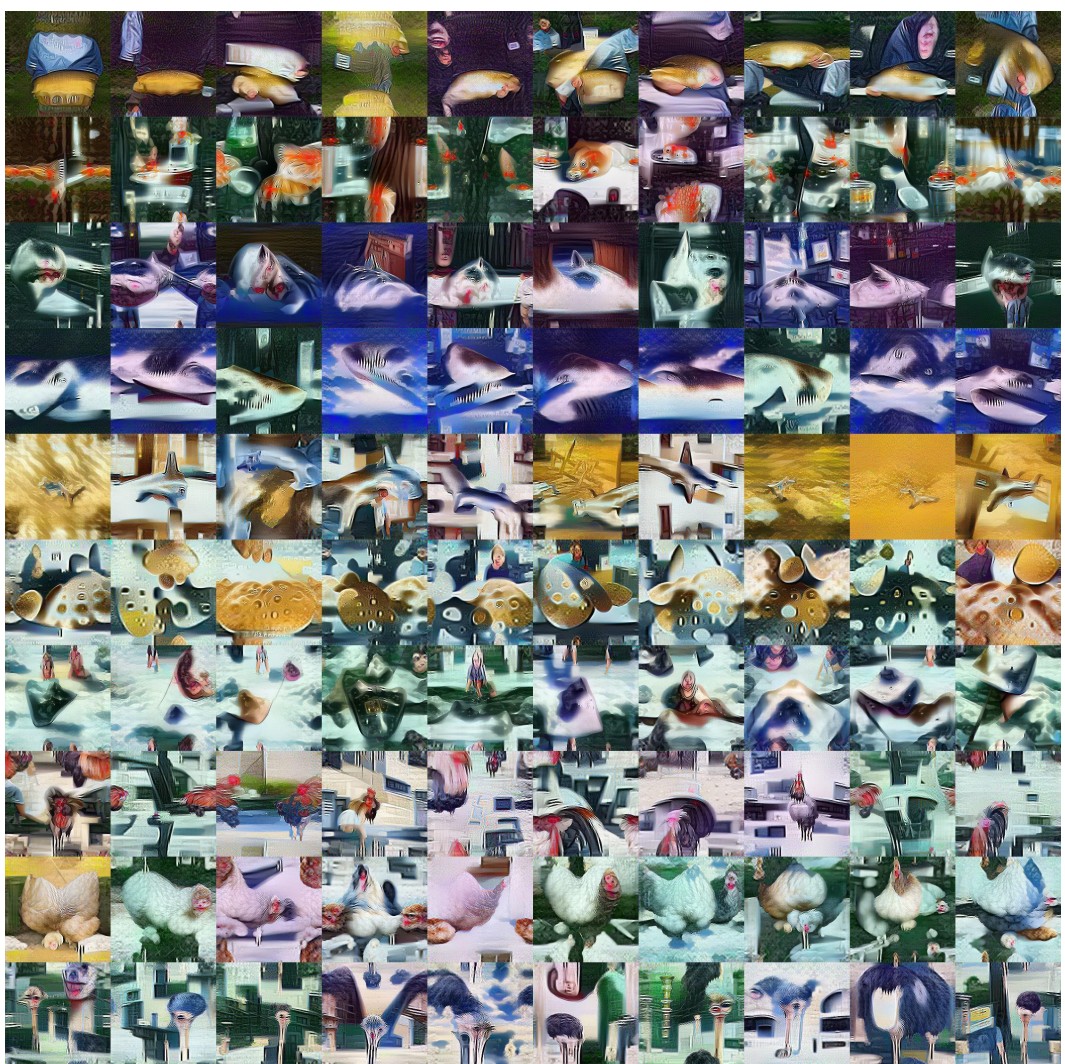

Figure 13: Visualization of top 10 classes in ImageNet-1K from CDA under IPC=100.

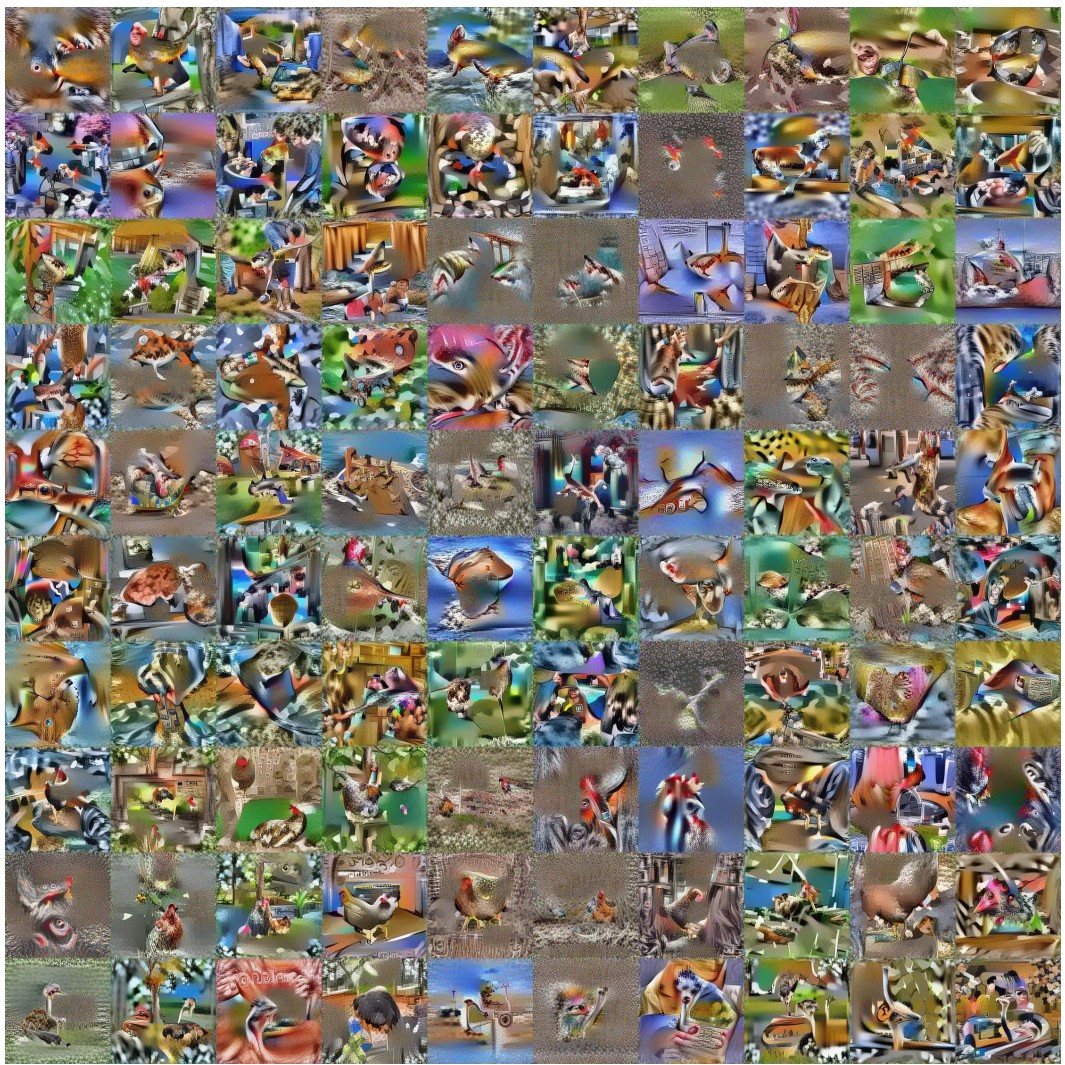

Figure 14: Visualization of top 10 classes in ImageNet-1K from G-VBSM under IPC=100.

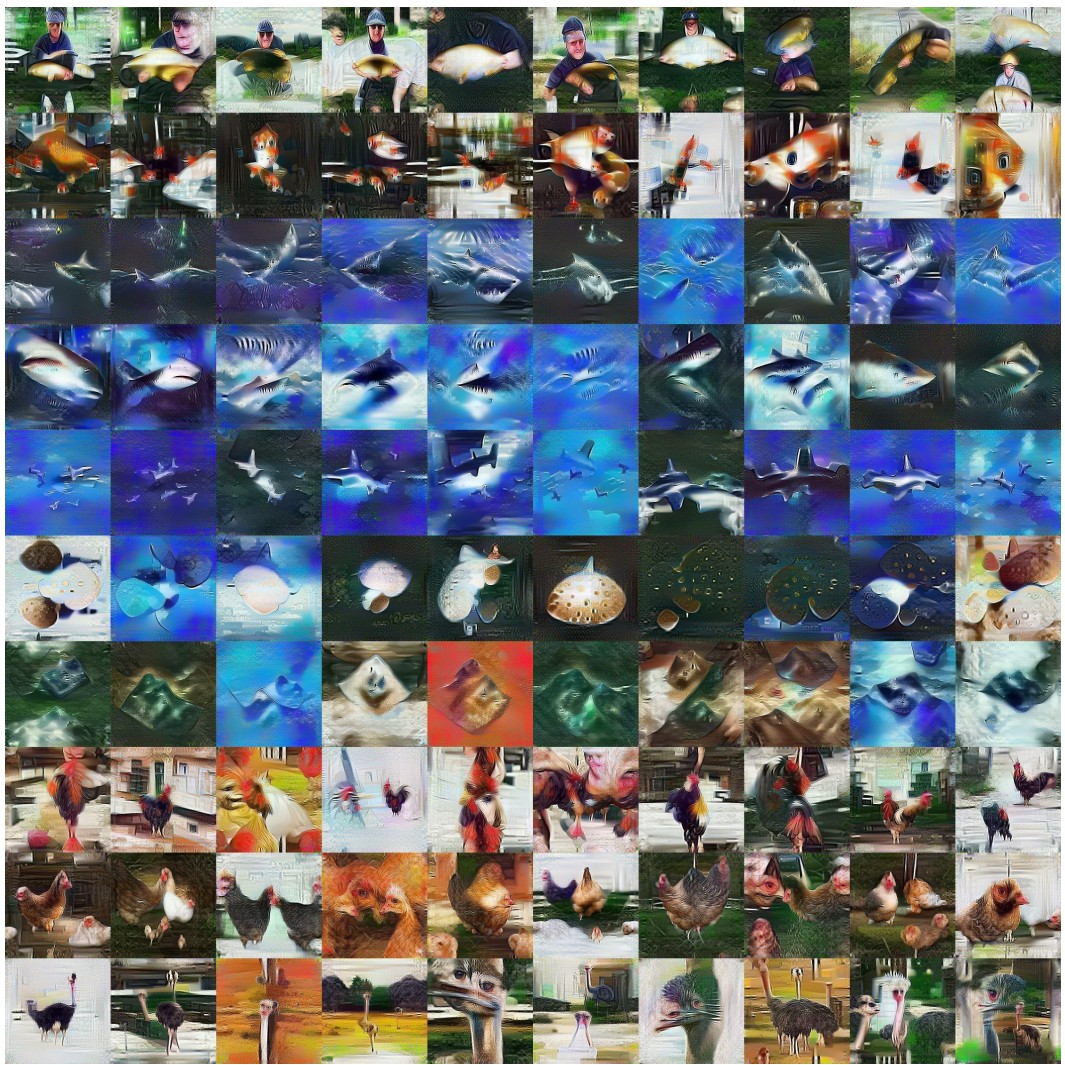

Figure 15: Visualization of top 10 classes in ImageNet-1K from DWA under IPC=100.

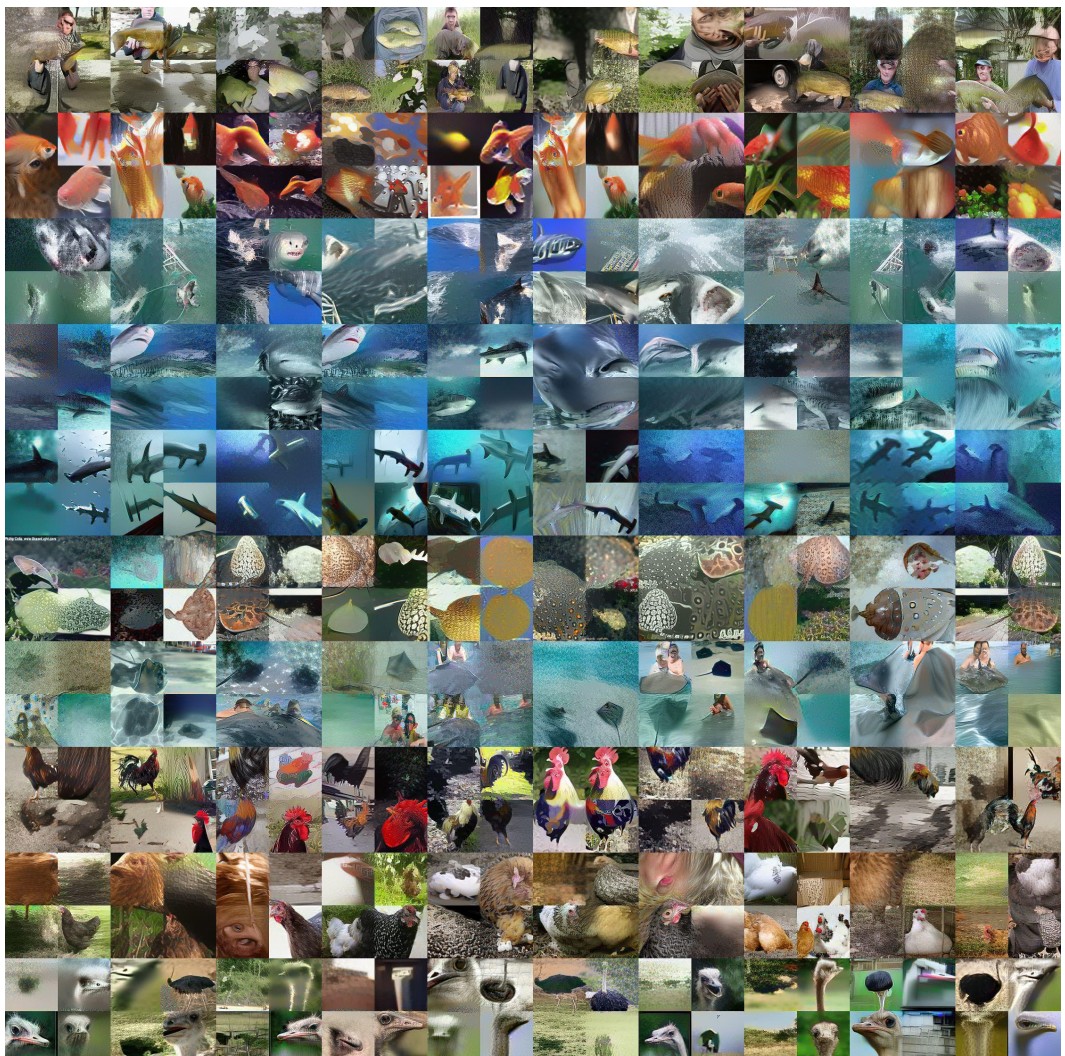

Figure 16: Visualization of top 10 classes in ImageNet-1K from EDC under IPC=100.

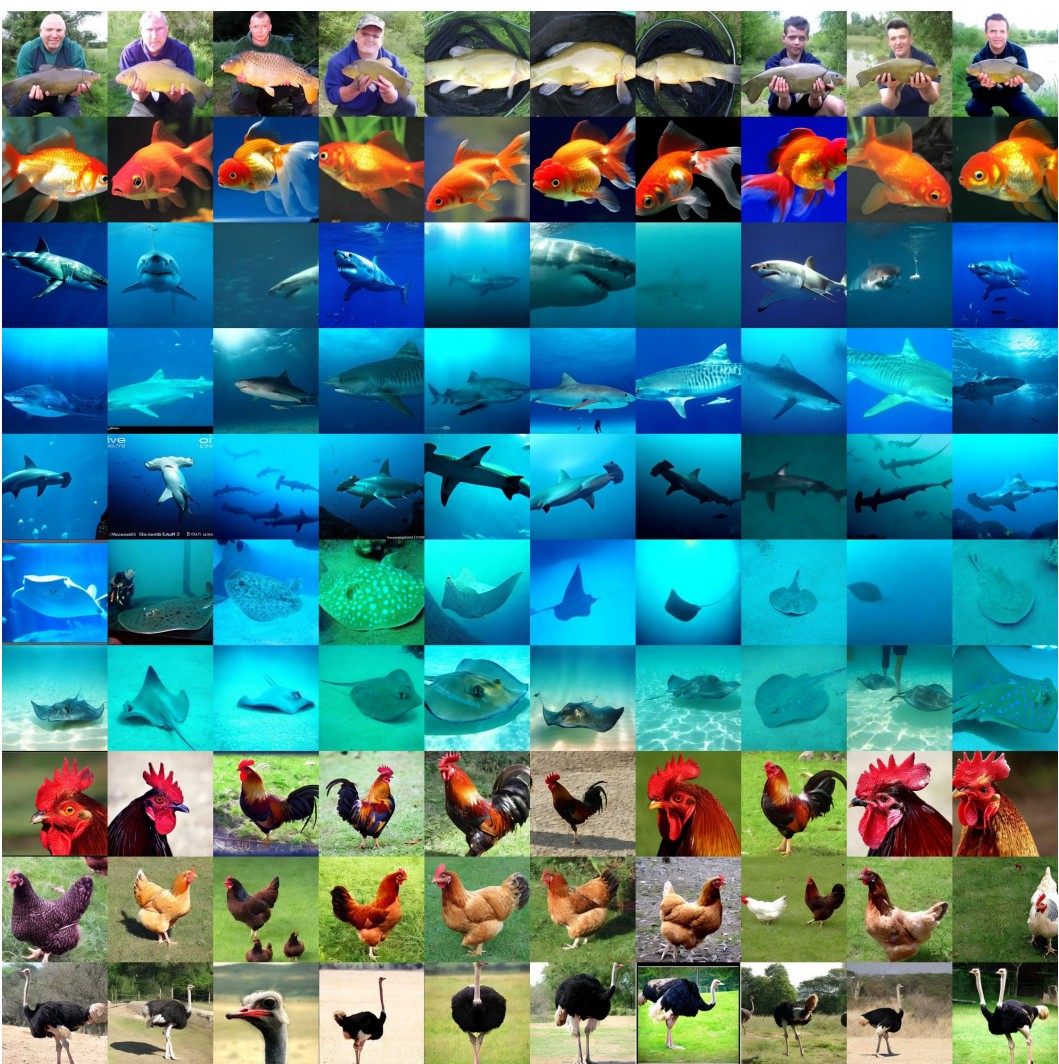

Figure 17: Visualization of top 10 classes in ImageNet-1K from Minimax under IPC=100.

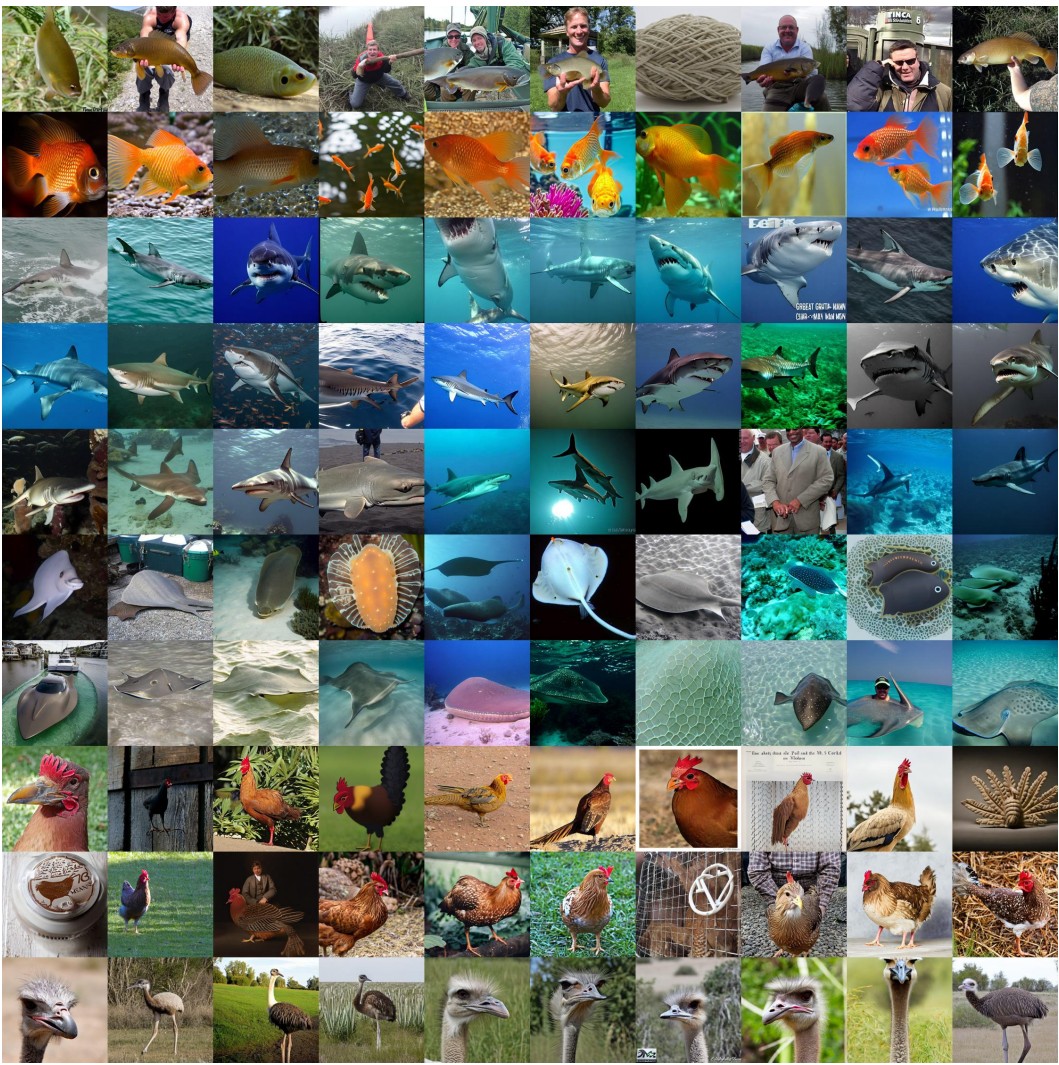

Figure 18: Visualization of top 10 classes in ImageNet-1K from $D^4M$ under IPC=100.

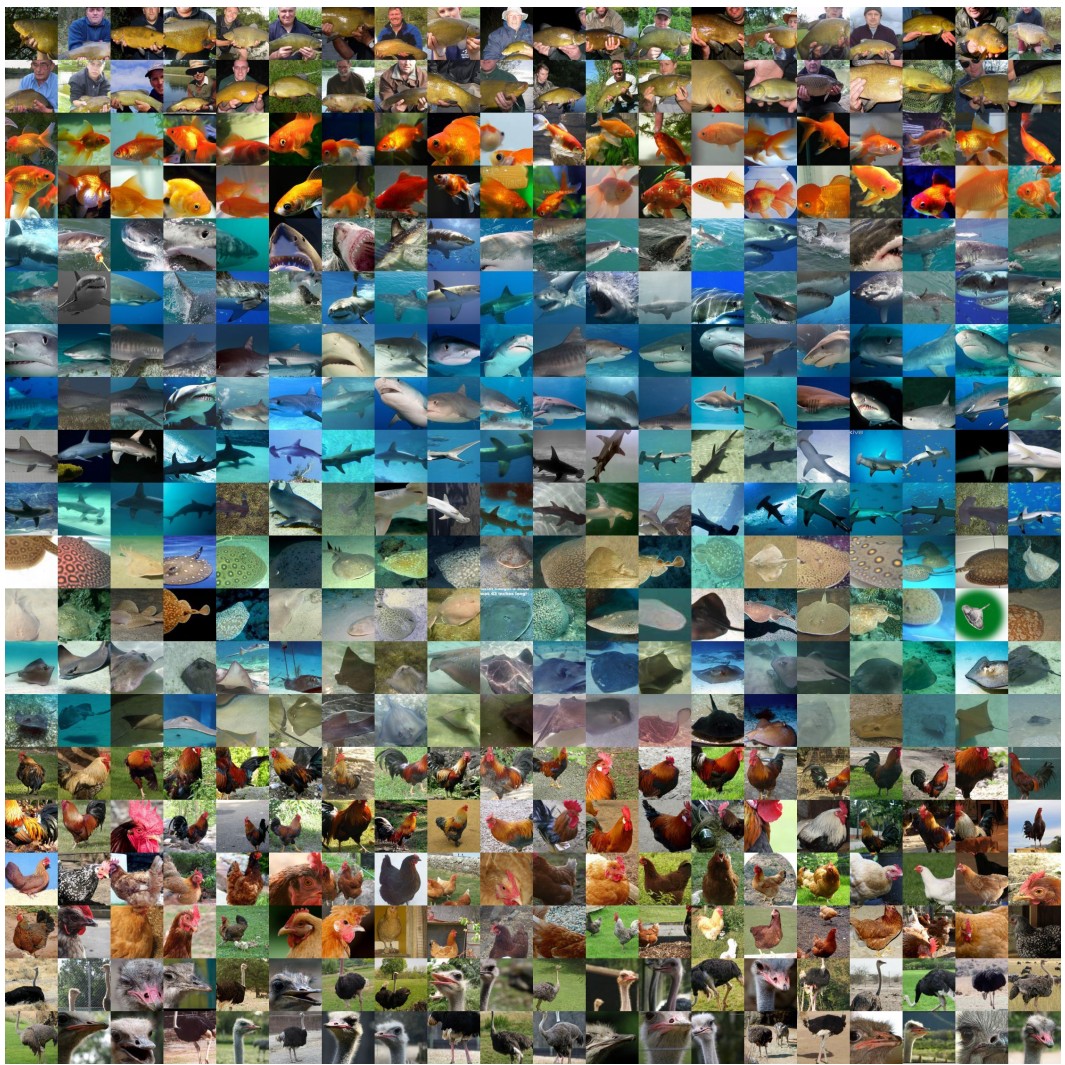

Figure 19: Visualization of top 10 classes in ImageNet-1K from RDED under IPC=100.

