# OpenReview forum: "Rectified Decoupled Dataset Distillation: A Closer Look for Fair and Comprehensive Evaluation"
_ICLR.cc/2026/Conference — ICLR 2026 Poster_

### Official Review · Reviewer_ihpb · 2025-10-15

**Soundness:** 4
**Presentation:** 3
**Contribution:** 3
**Rating:** 6
**Confidence:** 4

**Summary:**

This paper revisits the evaluation of *decoupled dataset distillation (DDD)* methods and introduces Rectified Decoupled Dataset Distillation (RD3), a unified benchmark framework for fair and comprehensive comparison.
The authors identify that prior DDD papers (e.g., SRe2L, CDA, DWA, EDC, G-VBSM, D4M) have used inconsistent post-evaluation settings—varying batch sizes, loss functions, schedulers, soft label configurations, and augmentations—making their reported performance incomparable.
RD3 standardizes all these evaluation aspects (batch size, loss, augmentations, optimization schedule, and model architectures), and re-evaluates the main DDD methods under identical conditions.
The results show that previously reported gaps of up to 27% shrink to ~6%, indicating that most gains in prior work were due to inconsistent settings rather than true algorithmic improvements.
Beyond unification, RD3 also provides diagnostic analyses, new metrics (efficiency, cross-architecture generalization), and an open-source benchmark for reproducible comparison.

**Strengths:**

* **Timely and valuable contribution:** RD3 addresses a real and growing problem in the dataset distillation community: inconsistent evaluation practices that lead to misleading comparisons.
* **Thorough analysis:** The paper systematically disentangles how batch size, loss function, learning rate scheduler, and soft-label design affect performance, providing empirical evidence that evaluation bias dominates algorithmic differences.
* **Unified and transparent benchmark:** All post-evaluation protocols are standardized, with clear rationale (e.g., KL loss for soft-label consistency; batch size = 50 as a fair trade-off).
* **Broader impact on reproducibility:** The open benchmark and diagnostic framework can serve as a long-term reference for fair evaluation of future DDD methods.
* **Clarity:** The paper is well-structured and transparent about design decisions and their effects.

**Weaknesses:**

* **Limited coverage of cutting-edge methods:**
  While RD3 re-evaluates several major DDD baselines, it currently does not include more *high-performing or near-lossless* distillation frameworks such as DATM and related trajectory-based optimization methods.
  Extending RD3 to evaluate these *lossless distillation* regimes would significantly strengthen its comprehensiveness and demonstrate that the framework remains valid even for next-generation algorithms that achieve nearly full-data performance.

* **Scope restricted to image datasets:**
  The current benchmark focuses solely on vision data. Enabling RD3 to handle other modalities—such as **graph distillation**  or multimodal datasets—would greatly expand its applicability and impact.

* **No new algorithmic contribution:**
  The contribution is primarily evaluative rather than methodological, which is appropriate for a benchmark paper but may limit its novelty for readers seeking new distillation algorithms.

**Questions:**

1. Could the authors integrate **trajectory-matching-based distillation methods**, to strengthen coverage and demonstrate RD3’s scalability to newer paradigms?
2. How feasible is it to extend RD3 to **non-image domains**, particularly graph data distillation (e.g., GCond[1], GEOM[2])? Would the same evaluation protocol and metrics apply?
3. Are there plans to host RD3 as a public leaderboard or benchmark platform?

[1] Graph condensation for graph neural networks. ICLR 2021.

[2] Navigating Complexity: Toward Lossless Graph Condensation via Expanding Window Matching. ICML 2024.

---

> ### Author Response · Authors · 2025-11-21
>
> >**Weakness 1 & Question 1: Integration of the  trajectory-matching-based distillation methods.**
>
> Thank you for your suggestion. The main reason we did not include trajectory matching-based methods is that these methods are not yet effectively deployable on ImageNet-1K due to the large time overhead required to obtain the pretrained trajectory weights. However, our proposed RD$^3$ framework can still evaluate such methods DATM [1] and EDF [2] on smaller datasets, including ImageNette and ImageWoof. The experimental results, shown in the table below, demonstrate the versatility of our proposed framework.
>
> |Method|IPC1|IPC10|IPC50|IPC100|
> |-|-|-|-|-|
> |RDED|31.4|63.8|86.8|89.6|
> |DATM|28.0|65.2|86.4|89.3|
> |EDF|28.4|68.2|88.1|90.5|
>
> |Method|IPC1|IPC10|IPC50|IPC100|
> |-|-|-|-|-|
> |RDED|20.3|46.5|72.0|78.6|
> |DATM|21.0|46.2|71.8|78.2|
> |EDF|21.5|47.4|72.9|79.8|
>
> >**Weakness 2 & Question 2: Extension to various modalities.**
>
> Thank you for your question. We appreciate the reviewer’s focus on other modalities. We recognize the growing interest in multimodal dataset distillation and graph distillation, however, **methods in these fields are still in the development stage and have not yet met the issue of the varying post-evaluation settings present in decoupled dataset distillation for image datasets**. Since the central idea of our work is to provide fair and unified evaluations, it can be easily extended to datasets from various modalities. The following aspects outline how our method aligns with various modalities dataset distillation:
>
> - Our work tackles the issue of unfairness in current decoupled dataset distillation evaluations. This issue is relevant to all modalities, as evaluation fairness depends on ensuring that post-evaluation settings are consistent across different methods.
>
> - We separates various factors, such as knowledge distillation and data augmentation, by comparing the improvements of different techniques and align all the methods. This approach can be easily adapted to other modalities as a universial workflow.
>
> As mentioned, our work will be continuously maintained. We will monitor the latest developments in dataset distillation across all domains and enhance our benchmark to ensure it remains comprehensive.
>
> >**Weakness 3: The organization of the manuscript.**
>
> Thank you for your valuable feedback. Our work focuses on addressing a critical issue in current decoupled dataset distillation research: the lack of standardized post-evaluation settings, which can lead to unpurified and potentially misleading performance gains. Through extensive and comprehensive experiments, we aim to raise awareness of this problem. As such, the contribution of our paper is primarily evaluative rather than methodological, aligning more with the style of a benchmark or position paper. While we do not propose a new algorithm, we believe our work provides essential insights and lays the groundwork for future research in distillation methods, offering a more reliable framework for evaluating existing techniques.
>
> >**Question 3: Plan of host RD$^3$ as a public leaderboard.**
>
> Thank you for your recognition. We will build a public leaderboard for our work in the future.

---

> > ### Comment · Reviewer_ihpb · 2025-11-23
> > **I thank the authors for the reply.**
> >
> > Regarding the debate on "fixed hyperparameters" raised by other reviewers, I acknowledge that a strictly fixed setting has limitations. To balance standardization and fairness, I suggest future iterations of RD3 consider a "Parameter Pool" approach (evaluating across a small set of common hyperparameters rather than a single fixed one) to verify robustness.
> >
> > However, given the paper's contribution in exposing the evaluation inconsistencies in the field, I believe its value outweighs these limitations. I maintain my score of 6.

---

> > > ### Author Response · Authors · 2025-11-23
> > >
> > > Thank you for your recognition. As discussed, our work has proposed a known optimal and universal post-evaluation setup, thus we have already completed the process of constructing a “parameter pool” and selecting the optimal settings for all methods. Because the performance of decoupled dataset distillation methods exhibits the same pattern of variation across all hyper-parameters, we highlight the need for a unified evaluation framework in this field. We will also provide all available hyper-parameter settings in future work.

---

### Official Review · Reviewer_94Vn · 2025-10-19

**Soundness:** 3
**Presentation:** 4
**Contribution:** 3
**Rating:** 6
**Confidence:** 3

**Summary:**

The paper proposes RD3, a standardized evaluation framework for decoupled dataset distillation. It argues and shows empirically that divergences in post-evaluation choices (batch size, LR schedule smoothing, data augmentations, label formulations, training epochs) drive a large share of performance differences across methods, not the synthesis procedure itself. The authors unify settings across optimization-based, selection-based, and generation-based approaches; evaluate on six datasets (CIFAR-10/100, TinyImageNet, ImageNet-1K, ImageNette/Woof); and test cross-architecture generalization (ResNet family, MobileNet-V2, EfficientNet-B0, Swin-T, ViT-B). They further emphasize efficiency (data generation time) and explore general techniques (alternative initialization, hybrid soft labels, MSE-GT loss) that materially shift outcomes under the same data.

**Strengths:**

1. **Clear empirical finding with real stakes for the field.** Under a consistent setup, the headline gap between methods shrinks dramatically (27.3% → 6.7% on ImageNet-1K @ IPC=10), reframing prior “SOTA” claims as largely protocol effects.


2.**Unified, well-documented evaluation protocol .** The paper spells out aligned epochs (400), batch size (BS=50 for R-18), smoothing cosine LR (ζ), and a fixed augmentation recipe, and shows incremental effects (Fig. 2). This is useful to both reproduce and compare.

3.The paper does a comphrhensive analysis and isolates factors many works quietly vary: initialization can help/hurt (e.g., noise harms DWA/EDC; RDED init boosts several methods), and hybrid soft labels help at low IPC regardless of method family.1

**Weaknesses:**

1.The augmentation suite is taken from RDED and applied universally. That is reasonable for standardization, but it may advantage methods whose images/labels interact best with that augmentation mix. A short sensitivity analysis (swap in SRe2L/CDA augmentations) would address perceived bias. Similarly, fixing BS=50 and ζ (e.g., ζ=1 for ResNets) is pragmatic, yet different families could prefer distinct regimes. The appendix defends ζ choices, but a small grid over {BS, ζ} across families would strengthen neutrality claims.

2.The efficiency plot excludes classifier training time for optimization/selection methods. I’d prefer an end-to-end accounting.

**Questions:**

1.For MSE-GT, you adopt γ from G-VBSM. Could you provide a small γ sweep (e.g., {0, 0.01, 0.025, 0.05}) across two families to verify that conclusions don’t hinge on one borrowed value?

2.In Appendix F, do you have analyses (e.g., attention maps, calibration, failure clusters) to explain the >40% ViT drop at IPC=10 and why it recovers at higher IPC? Any indications that label entropy or image diversity drives the effect?

3.In Table1, you standardize on a single ResNet-18 teacher for soft labels. What happens if the teacher is ViT-B or an ensemble (distinct from hybrid labels used at relabel time)? I am curious about different teacher models' effects. (I checked the result in Appendix F, but it doesn’t test ViT-B or ensembles as the teacher)

---

> ### Author Response · Authors · 2025-11-21
>
> >**Weakness 1: More ablation study on hyper-parameters.**
>
> Thank you for your suggestion. Actually our setting is not entirely consistent with RDED. For batch size and the learning rate scheduler, we have chosen more optimal settings, as described in **Section 3.2** and **Table 2**. Additionally, we present more performance improvement techniques in **Sections 4.3** and **Section 4.4**. Following your suggestion, we have added additional ablation experiments using the RD$^3$ hyperparameters to ensure the fairness of the framework we propose. In **Figure 2**, we present the performance changes of different categories of methods as our proposed evaluation settings are progressively applied. Additionally, we provide the performance variations of all methods under this gradual setting in the table below.
>
> |Config|SRe$^2$L|CDA|G-VBSM|DWA|EDC|Minimax|D$^4$M|RDED|
> |-|-|-|-|-|-|-|-|-|
> |Simplified|26.6|27.2|27.1|28.7|31.3|30.7|30.5|30.9|
> |+Batchsize|35.2|36.4|36.9|37.7|41.2|40.4|40.7|41.2|
> |+Data Augmentation|38.3|39.4|39.7|40.6|43.1|42.7|42.5|43.9|
> |+LRS ($\zeta$)|40.2|41.2|41.5|42.5|46.9|45.9|45.4|46.3|
>
> We provide the ablation experiments for all data augmentation operations with split and cumulative settings, as shown in the table below.
>
> |Config|SRe$^2$L|CDA|G-VBSM|DWA|EDC|Minimax|D$^4$M|RDED|
> |-|-|-|-|-|-|-|-|-|
> |None|2.7|3.8|4.4|5.6|7.4|6.7|5.4|7.2|
> |+CutMix|18.4|20.5|21.2|23.2|31.0|28.9|27.4|30.1|
> |+RandomCrop|37.2|38.8|39.2|40.1|44.7|43.2|42.5|43.1|
> |+RandomFlip|38.9|40.3|40.5|41.3|45.7|44.7|44.0|45.2|
> |+Patch Shuffle|40.2|41.2|41.5|42.5|46.9|45.9|45.4|46.3|
>
>
>
> In addition to the ablation for the smoothing factor $\zeta$ presented in **Figure 4**, we also include ablation experiments using different batchsize in the table below.
>
> |Batchsize|SRe$^2$L|CDA|G-VBSM|DWA|EDC|Minimax|D$^4$M|RDED|
> |-|-|-|-|-|-|-|-|-|
> |1024|26.6|27.2|27.1|28.7|31.3|30.7|30.5|30.9|
> |512|33.2|34.1|34.8|36.3|39.0|38.2|38.6|39.1|
> |256|37.4|38.2|38.7|39.9|42.5|41.8|42.3|42.0|
> |50|40.2|41.2|41.5|42.5|46.9|45.9|45.4|46.3|
>
> >**Weakness 2: The training time of the classifiers.**
>
> Since all decoupled dataset distillation methods require the use of soft labels as an auxiliary, and given that pre-trained classifier weights for ImageNet-1K are publicly available, we have excluded the time spent on training the classifier. Under the same experimental setup, the classifier training time for different methods on ImageNet-1K are shown in the table below. G-VBSM and EDC, due to the use of multiple teacher models, have longer classifier training times.
>
> |SRe$^2$L|CDA|G-VBSM|DWA|EDC|Minimax|D$^4$M|RDED|
> |-|-|-|-|-|-|-|-|
> |160h|160h|700h|160h|620h|160h|160h|160h|

---

> ### Author Response · Authors · 2025-11-21
>
> >**Question 1: More ablation study on $\gamma$.**
>
> We provide additional ablation experiments for $\gamma$, as shown in the table below. It can be observed that for different values of $\gamma$, all methods exhibit the same pattern of performance changes, demonstrating that we have correctly identified techniques that have a universal effect across all categories of decoupled distillation methods.
>
> |$\gamma$|SRe$^2$L|CDA|G-VBSM|DWA|EDC|Minimax|D$^4$M|RDED|
> |-|-|-|-|-|-|-|-|-|
> |0|36.3|37.4|37.2|38.8|41.7|42.0|41.0|43.1|
> |0.01|38.8|39.6|40.2|41.1|45.4|44.1|43.5|42.0|
> |0.025|40.2|41.2|41.5|42.5|46.9|45.9|45.4|46.3|
> |0.05|40.4|40.8|41.6|42.4|46.5|46.3|45.2|45.9|
>
> >**Question 2: Experimental phenomenon on ViT-based models.**
>
> Thank you for your suggestion. In fact, it is a well-established conclusion that transformer-based models typically require more training data than CNN-based models to achieve better performance. We have found that this conclusion also applies to distilled datasets. This same experimental observation has been noted in existing works [1] [2] [3]. We will include some qualitative visualization results in the final version, and we look forward to further discussion with you.
>
>
> >**Question 3: Experimental results of using different teacher network.**
>
> In the main experimental setup, we use a single pre-trained ResNet-18 as the teacher model to generate soft labels. In **Table 5**, we present the results of using multiple teacher models, namely ResNet-18, MobileNet-V2, AlexNet, and EfficientNet-B0, to generate mixed soft labels by averaging the outputs. The experimental results, shown in the table below, demonstrate that using different teacher models leads to significant performance variations. This highlights the necessity of the unified evaluation setup we propose, which incorporates the use of soft labels.
>
> |Method|ResNet-18|ResNet-50|ResNet-101|MobileNet-V2|EfficientNet-B0|Swin-V2-T|ViT-B-16|
> |-|-|-|-|-|-|-|-|
> |SRe$^2$L|55.2|48.3|46.4|51.3|42.2|29.4|26.3|
> |CDA|56.7|49.2|47.3|52.9|42.4|31.0|27.1|
> |G-VBSM|56.6|48.5|46.8|53.2|43.7|30.5|27.8|
> |DWA|57.7|50.3|48.1|54.1|44.2|32.0|28.7|
> |EDC|60.1|52.2|50.3|56.4|46.8|34.7|31.3|
> |Minimax|60.4|52.6|50.1|56.4|47.3|35.2|32.0|
> |D$^4$M|60.2|51.4|50.6|55.8|47.0|35.1|30.9|
> |RDED|58.9|51.0|49.4|55.7|45.8|32.5|29.5|
>
> [1] Generalized Large-Scale Data Condensation via Various Backbone and Statistical Matching. In CVPR, 2024.
>
> [2] D4M: Dataset Distillation via Disentangled Diffusion Model. In CVPR, 2024.
>
> [3] Towards Model-Agnostic Dataset Condensation by Heterogeneous Models. In ECCV, 2024.

---

> > ### Comment · Reviewer_94Vn · 2025-11-25
> > **Thanks the authors for provding more experimental results.**
> >
> > With the new provided results, most of my concerns are clearly addressed. According to the paper propose a unified and consistant evaluation protocol for the field, that is a promising contribution, I will raise my score to 8.

---

> > > ### Author Response · Authors · 2025-11-25
> > >
> > > Thank you for your recognition of our work. We hope that our work can provide a fair comparison environment and new insights for the field of dataset distillation. If you have any further questions, we would be happy to engage in discussion with you.

---

### Official Review · Reviewer_McZF · 2025-11-01

**Soundness:** 2
**Presentation:** 2
**Contribution:** 3
**Rating:** 4
**Confidence:** 2

**Summary:**

The paper argues that evaluation across existing dataset distillation methods are inconsistent. In consequence, it is unclear whether performance gains from new methods is due to improvement from the distillation mechanism or auxiliary improvement in the downstream model training. To overcome this, the paper introduces Rectified Decoupled Dataset Distillation (RD3) to systematically investigate how different evaluation settings affect test accuracy. RD3 find that previous observed performance increase of over 27% reduces to under 7% with the same evaluation setting.

**Strengths:**

1. The paper point out a concerning trend with current dataset distillation evaluation protocol that can misrepresent the capabilities of existing algorithm.
2. The paper provides a rather thorough analysis towards understanding the capabilities of current dataset distillation methods, revealing the performance gap between the baseline SRe2L and RDED is a lot smaller than expected.
3. To the best of my knowledge, this is the first work that provide comprehensive training time, generalization, and initialization analysis, which are important considerations for people that use these dataset distillation algorithm in practice.

**Weaknesses:**

1. The problem of different evaluation procedure exist is many subfields of machine learning and not unique to dataset distillation. More discussion on the unique challenges that exist in dataset distillation would strength the overall motivation of the paper.
2. The presentation of many arguments in the paper can substantially be improved (see question 1-3).
3. Minor: Figure 3 can substantially be improved. There are eight points on the graph that differs only in color. This makes it difficult to read and not color-blind friendly.

**Questions:**

1. The paper argues "to ensure fairness, all methods should adopt a unified training setting, regardless of their specific motivations" but it is unclear from the paper why this matters or ensures fairness. Why does using the same evaluation procedure enforce fairness? For instance, method A, compared to method B, can benefit more in from training procedure X than Y. Is it fair to evaluate with procedure X or Y? To a partitioner, the will only care about the final performance on the downstream task.
2. Figure 2 argues that different techniques leads to different performance impact by why is this the case? The accuracy gains from the technique isn't the same but the relatively ranking between the two method is consistent with the augmentations. Hence, the same conclusion can be drawn.
3. Section 4.2 argues "in comparison to marginal differences in test accuracy, computational efficiency should be considered a more critical criterion for evaluating different methods". What is considered marginal? For example, RDED takes 1 hour to train and achieve 61.5% but D4M takes 52 hours to train and achieve 63.2%. To a practitioner, the 1-2% improvement in accuracy maybe well worth the 50x increase in training time.
4. What is the total storage size (including soft labels) across the different methods?
5. How does this differ from DD-Ranking?

Li, Zekai, et al. "Dd-ranking: Rethinking the evaluation of dataset distillation." arXiv preprint arXiv:2505.13300 (2025).

---

> ### Author Response · Authors · 2025-11-21
>
> >**Weakness 1: Discussion on the challenge in dataset distillation.**
>
> In most deep learning fields, only model training is required, whereas dataset distillation tasks involve two stages: training the distilled dataset and then using it to train and evaluate the model. This introduces unique and more implicit challenges. For instance, the performance improvements in some decoupled dataset distillation methods stem from optimizing the loss function rather than optimizing the dataset itself, as shown in **Table 6**. This is inconsistent with the majority of deep learning fields, where optimizing the loss function is generally accepted.
>
> Additionally, we re-evaluated the performance of existing methods, providing the true performance compared to the original papers, as shown in **Table 2**. Our work offers a fair and thorough performance comparison for future dataset distillation methods, advancing the positive development of the dataset distillation field.
>
> >**Weakness 2 & Question 1: Clarification of using the unified setting.**
>
> Existing decoupled dataset distillation methods assume that **Method A** is evaluated under **setting X**, while **Method B** is evaluated under **setting Y**, resulting in better performance for **Method B**, which is then reported. **Our work aims to highlight that Method A should be re-evaluated under setting Y to ensure that the performance improvement truly comes from the enhancement in dataset quality**, a core concern of the dataset distillation field, rather than merely using a more favorable post-evaluation setting.
>
> As shown in **Figure 1**, the performance improvement between different methods are significantly reduced when the same evaluation settings are used, demonstrating the existence of the issue we discuss. Additionally, in **Table 2**, we provide the performance reported in both our framework and the original papers. It can be seen that our method achieves consistent performance improvements across all compression rate settings except for EDC, demonstrating that our framework represents the optimal known **setting Y**. Therefore, there is no evidence that our use of a less favorable evaluation setting suppresses the performance of certain methods. The reasons for the performance decline in EDC are also explained in the data provided in **Tables 5** and **Table 6**.
>
>
>
> >**Weakness 2 & Question 2: Clarification of the technique performance.**
>
> **The point we aim to demonstrate with Figure 2 is not that different techniques lead to different performance improvements, but rather that techniques can enhance final performance without altering the distilled dataset**. In fact, the post-evaluation settings and corresponding performance of SRe$^2$L's original paper are shown in the "simplified" column of **Figure 2**, while the post-evaluation settings and related performance of RDED's original paper are shown in the "Epoch" column. **It is evident that after aligning the evaluation settings, the performance gap between the two methods is dramatically reduced**. This consistent phenomenon is also observed in other methods, as shown in **Figure 1**, further demonstrating the necessity of our work in clarifying the dataset distillation field.
>
>
>
>
>
> >**Weakness 2 & Question 3: Clarification of selecting training cost as evaluation metric.**
>
> We acknowledge the improvement in absolute performance; however, in the downstream areas of dataset distillation, such as continual learning, computational overhead also plays a crucial role, and it has been a key consideration in many existing dataset distillation works [1] [2] [3] [4] [5].  Therefore, after re-evaluating the true performance of existing methods, such as RDED, G-VBSM, and Minimax, we observe relatively small or even zero performance gains. In contrast, the computational overhead can be tens or even hundreds of times greater. **This substantial difference in computational cost should be recognized by future work as an important consideration and used as an evaluation metric, providing a new perspective for more comprehensive assessments**.
>
> [1] Going Beyond Feature Similarity: Effective Dataset Distillation based on Class-aware Conditional Mutual Information. In ICLR, 2025.
>
> [2] Dataset Distillation with Neural Characteristic Function: A Minmax Perspective. In CVPR, 2025.
>
> [3] Beyond Random: Automatic Inner-loop Optimization in Dataset Distillation. In NeurIPS, 2025.
>
> [4] GIFT: Unlocking Full Potential of Labels in Distilled Dataset at Near-zero Cost. In ICLR, 2025.
>
> [5] FADRM: Fast and Accurate Data Residual Matching for Dataset Distillation. In NeurIPS, 2025.

---

> ### Author Response · Authors · 2025-11-21
>
> >**Weakness 3: Adjustment of the figure.**
>
> Thank you for your suggestion. We will use different icons for each method in **Figure 3** in the final version.
>
>
> >**Question 4: Total storage size of distilled datasets.**
>
> Since all methods use the same evaluation setting, the storage overhead is identical. In the table below, we present the storage space required for images with different compression rates and soft labels on ImageNet-1K.
>
> ||IPC1|IPC10|IPC50|IPC100|
> |-|-|-|-|-|
> |Image|0.19G|0.37G|0.75G|1.49G|
> |Label|7.08G|14.17G|28.33G|56.66G|
>
>
> >**Question 5: The difference between our work and related works.**
>
> DD-Ranking aims to propose a new, universally applicable metric for achieving relatively fair comparisons across different category methods. In contrast, our work focuses on test accuracy as the decisive metric, highlighting the factors that influence the post-evaluation settings of decoupled datasets and purifying the true performance improvements of existing methods, as shown in **Figure 1** and **Table 2**. We provide a new, detailed, and fair benchmark. Therefore, our work is orthogonal to and does not overlap with DD-Ranking. Further clarification is provided in **Appendix A.3**.

---

> > ### Comment · Reviewer_McZF · 2025-11-26
> >
> > Thank you for the response. My concerns are addressed and I will be raising my score.

---

> > > ### Author Response · Authors · 2025-11-26
> > >
> > > We are very delighted to see that your concerns have been addressed. We appreciate the support. Your feedback is very helpful for us to improve our work, and we will incorporate the additional discussion in the revision based on your advice. If there are any further questions, don't hesitate to post here for discussion. Thank you.

---

### Official Review · Reviewer_7EFG · 2025-11-03

**Soundness:** 2
**Presentation:** 2
**Contribution:** 2
**Rating:** 2
**Confidence:** 2

**Summary:**

This paper introduces a unified and systematic evaluation framework for dataset distillation to compare decoupled distillation methods.

**Strengths:**

1. The evaluation is systematic and practical, providing valuable guidance for empirical method comparison and design.

**Weaknesses:**

1. Lacks fundamental analysis—most observations are based on empirical results and hypotheses.
2. Uses a fixed hyper-parameter set for all methods. As an evaluation framework, this approach is unconvincing since different methods may have different optimal hyperparameters.
3. Some design choices require additional justification (see questions below).

**Questions:**

1. Should convergence speed be considered a characteristic of a method? If some methods converge faster than others, using longer training epochs may negatively affect their performance.
2. The cross-architecture results are interesting. Would replacing the soft label network with a different network change the results?

---

> ### Author Response · Authors · 2025-11-21
>
> >**Weakness 1: Lack of the fundamental analysis.**
>
> Our work addresses a critical issue in current decoupled dataset distillation research: the lack of standardized post-evaluation settings, which can lead to unpurified and potentially misleading performance gains. While much of the discussion is based on empirical results and hypotheses, we provide extensive and comprehensive experiments to underscore this concern. Given the nature of our analysis, the writing style of this paper aligns more closely with that of a benchmark or position paper.
>
>
>
>
> >**Weakness 2: Fixed hyper-parameter setting for all methods.**
>
> Our method aims to propose a general framework for fair comparisons, thus it is essential to perform a unified evaluation using the same hyperparameter settings, which is widely recognized in other fields, such as knowledge distillation. Furthermore, we acknowledge that subsequent work may achieve higher performance by adjusting hyperparameters. However, **our work emphasizes that any performance improvement gained through hyperparameter tuning needs to be verified against existing baseline methods**, as outlined in lines 79-81. This point is also demonstrated in **Figure 1**.
>
>
>
>
>
>
> >**Weakness 3 & Question 1: Consideration of convergence speed.**
>
> As discussed in **Section 3.2** regarding the number of training epochs, existing methods that converge quickly use 300 iterations as the post-evaluation setting. However, in our preliminary experiments, we found that increasing the number of training iterations for certain methods can effectively improve performance, as shown in **Figure 2**. We acknowledge that methods with faster convergence offer better computational efficiency, but this needs further clarification and should be decoupled from the performance metric of test accuracy.
>
> Therefore, our work adopts 400 epochs as a unified optimization setting, balancing both time efficiency and fairness. Additionally, we provide visualizations of the loss functions and related qualitative analysis for different methods during the post-training process in **Appendix P.1**.
>
>
>
>
>
> >**Weakness 3 & Question 2: Experimental results of using different teacher network.**
>
> In the main experimental setting, we use a single pre-trained ResNet-18 as the teacher model to generate soft labels. In **Table 5**, we show the results of using multiple teacher models to generate mixed soft labels, averaging the soft labels produced by ResNet-18, MobileNet-V2, AlexNet, and EfficientNet-B0. We provide the experimental results of using different teacher models to generate soft labels on ImageNet-1K, as shown in the table below. The use of different teacher models leads to significant performance changes, demonstrating the necessity of the unified evaluation setup we propose, which includes the use of soft labels.
>
> |Method|ResNet-18|ResNet-50|ResNet-101|MobileNet-V2|EfficientNet-B0|Swin-V2-T|ViT-B-16|
> |-|-|-|-|-|-|-|-|
> |SRe$^2$L|55.2|48.3|46.4|51.3|42.2|29.4|26.3|
> |CDA|56.7|49.2|47.3|52.9|42.4|31.0|27.1|
> |G-VBSM|56.6|48.5|46.8|53.2|43.7|30.5|27.8|
> |DWA|57.7|50.3|48.1|54.1|44.2|32.0|28.7|
> |EDC|60.1|52.2|50.3|56.4|46.8|34.7|31.3|
> |Minimax|60.4|52.6|50.1|56.4|47.3|35.2|32.0|
> |D$^4$M|60.2|51.4|50.6|55.8|47.0|35.1|30.9|
> |RDED|58.9|51.0|49.4|55.7|45.8|32.5|29.5|

---

> ### Comment · Reviewer_7EFG · 2025-11-22
>
> Thanks for the response and the additional experiments!
>
> Re:
>
> Weakness 1
>
> I agree that the comparison issue is crucial for developing and deepening our understanding of this domain. However, the current results do not clearly reveal what the essential ingredients are that make the method work. If I overlooked them, could the authors summarize the **additional insights** revealed by the unified benchmark?
>
> Weaknesses 2
>
> One fixed Hyperparameters set does not equal the Same Parameter but a fair comparison. To give a fair comparison and as a benchmark, we should test the method in a given search space (or given search budget) just like [DomainBed](https://github.com/facebookresearch/DomainBed).
>
>
> Weakness 3
>
> The statement “our work adopts 400 epochs as a unified optimization setting” raises a similar fairness concern. Fixing the number of epochs for all methods does not yield a fair comparison because different models converge at different rates. Instead, each method should be trained under multiple epoch budgets, with performance reported based on a principled selection criterion (e.g., validation performance). As the authors themselves note, some models may converge faster—would their performance degrade or plateau if forced to train substantially longer than necessary?

---

> ### Author Response · Authors · 2025-11-22
>
> We must explicitly point out that existing decoupled dataset distillation works have used different post-evaluation settings for comparisons, **whereas our method demonstrates that a more optimal post-evaluation setup is method-independent rather than sensitive to specific methods**. We provide various method-independent performance improvement techniques and purify performance differences. For instance, EDC improves performance across all settings to achieve very high results; however, **during performance comparisons, the baseline method used still relies on suboptimal evaluation settings rather than their optimal settings, making such comparisons completely unfair**.
>
> **In other areas of deep learning, different methods may achieve optimal performance under different settings. However, our work challenges and breaks this bias in decoupled dataset distillation**. Additionally, in **Table 2**, we present the performance of all methods under both the original post-evaluation settings and our framework. As shown, all methods, except for EDC under specific compression rates, achieve significant performance improvements and SOTA performance. This is because **our framework has already completed the search process you mentioned, identifying a universal optimal post-evaluation setup for all methods**.
>
> For the number of training epoch, we provide ablation experiments in the table below, where a consistent performance change pattern is observed across all methods. It is important to note that in the original work, aside from EDC using a different number of training epoch under specific setting, all other methods used 300 as the unified setting. Therefore, we follow the standard practice in the field, only modifying the specific value.
>
> |Epoch|SRe$^2$L|CDA|G-VBSM|DWA|EDC|Minimax|D$^4$M|RDED|
> |-|-|-|-|-|-|-|-|-|
> |300|39.1|40.0|40.1|41.3|46.0|44.2|44.3|45.6|
> |400|40.2|41.2|41.5|42.5|46.9|45.9|45.4|46.3|
>
> **It is worth emphasizing that our work is aimed at clarifying that performance comparison should eliminate the influence of evaluation settings**. This does not contradict your point about finding the optimal setting for different methods, **as the current optimal setting we have identified for the decoupled dataset distillation field is universal**. This demonstrates that the large performance differences observed between methods, as shown in **Figure 1**, actually arise from the lack of consistency in post-evaluation settings.
>
> As stated in **Section 1**, we acknowledge that different methods may use specific post-evaluation settings. However, **such modification need to be validated on the other baselines to verify their applicability and specificity**. This is precisely the issue that our work points out and addresses.

---

> ### Author Response · Authors · 2025-11-27
>
> Dear Reviewer,
>
> I hope this message finds you well. As the discussion period is nearing its end with less than five days remaining, I wanted to ensure we have addressed all your concerns satisfactorily. If there are any additional points or feedback you'd like us to consider, please let us know. Your insights are invaluable to us, and we're eager to address any remaining issues to improve our work.
>
> Thank you for your time and effort in reviewing our paper.

---

### Meta-Review · Area_Chair_RxG3 · 2026-01-05

**Summary:**

Reviewers generally agree that the paper addresses an important issue in dataset distillation: inconsistent post-evaluation protocols can misrepresent algorithm performance. The systematic analysis, unified evaluation setup, and the empirical finding that performance gaps shrink under consistent protocols are viewed as valuable contributions, with positive implications for reproducibility.

Key concerns include limited methodological novelty, as the work is primarily evaluative rather than proposing new algorithms, as well as potential bias introduced by fixing hyperparameters, augmentation schemes, and batch sizes that may favor certain methods. Reviewers also note that some claims rely heavily on empirical observations without full theoretical justification, and that the evaluation scope is restricted to vision datasets, excluding some recent high-performing or trajectory-based methods. Minor issues related to presentation clarity and incomplete efficiency accounting are also mentioned.

Overall, reviewers acknowledge the relevance and rigor of the study but differ on whether its novelty and scope are sufficient for acceptance. As a result, forming a clear meta-review decision is challenging. However, after carefully reading the paper and the authors’ responses, I find the work to be reasonably complete and insightful, and therefore recommend acceptance as a poster.

**Reviewer Concerns:**

Several reviewer concerns were substantially addressed during the rebuttal through additional experiments and clarifications. In particular, concerns regarding the fairness of adopting a unified post-evaluation protocol were directly addressed with new empirical evidence, which improved reviewer confidence. The authors also added ablation studies on key hyperparameters (e.g., batch size, augmentation, optimizers, loss functions, and teacher models), alleviating concerns about potential bias from fixed evaluation settings. Questions about the generalizability of the observed improvements and the inclusion of computational efficiency as an evaluation criterion were also clarified.

However, some concerns remain partially outstanding. The work continues to rely primarily on empirical analysis, with limited theoretical grounding. Overall, despite these remaining limitations, the strengths of the paper outweigh its weaknesses, and I believe that most major concerns have been adequately addressed.

**Reviewer Scores:**

Reviewer  7EFG:
Given the additional experiments and clarifications in the rebuttal, some concerns regarding fixed hyperparameters and evaluation fairness would likely be partially alleviated. However, due to the reviewer’s emphasis on methodological depth, I expect the reviewer would remain negative overall, with at most a modest score increase.

Reviewer McZF:
With full participation in the discussion, the score would likely increase slightly.

Reviewer 94Vn:
Given the added ablations and clarifications, most concerns would be addressed. I expect the reviewer would maintain a marginally positive score.

Reviewer ihpb:
This reviewer was generally supportive, with concerns mainly about scope and novelty. As these issues are largely unchanged, the score would likely remain the same.

---

### Decision · Program_Chairs · 2026-01-26

Accept (Poster)